EMBO
Molecular Medicine

# Uncovering a conserved vulnerability site in SARS-CoV-2 by a human antibody

Tingting Li[1,2,†] (iD), Hongmin Cai[1,2,†], Yapei Zhao[2,3,†] (iD), Yanfang Li[1,2,†], Yanling Lai[1,2,†], Hebang Yao[1,2,†] (iD), Liu Daisy Liu[1,2,†], Zhou Sun[4,†], Martje Fentener van Vlissingen[5,6] (iD), Thijs Kuiken[6,7], Corine H GeurtsvanKessel[6,7], Ning Zhang[8], Bingjie Zhou[2,3], Lu Lu[2,3], Yuhuan Gong[2,8], Wenming Qin[9], Moumita Mondal[3,10], Bowen Duan[2,3], Shiqi Xu[2,3] (iD), Audrey S Richard[6], Hervé Raoul[6], JianFeng Chen[1], Chenqi Xu[1], Ligang Wu[1], Haisheng Zhou[1,2], Zhong Huang[3,10] (iD), Xuechao Zhang[4], Jun Li[4], Yanyan Wang[1,2], Yuhai Bi[2,8], Barry Rockx[6,7], Junfang Chen[4,*] (iD), Fei-Long Meng[1,2,**] (iD), Dimitri Lavillette[3,10,11,***] (iD) & Dianfan Li[1,****] (iD)

## Abstract

An essential step for SARS-CoV-2 infection is the attachment to the host cell receptor by its Spike receptor-binding domain (RBD). Most of the existing RBD-targeting neutralizing antibodies block the receptor-binding motif (RBM), a mutable region with the potential to generate neutralization escape mutants. Here, we isolated and structurally characterized a non-RBM-targeting monoclonal antibody (FD20) from convalescent patients. FD20 engages the RBD at an epitope distal to the RBM with a $K_D$ of 5.6 nM, neutralizes SARS-CoV-2 including the current Variants of Concern such as B.1.1.7, B.1.351, P.1, and B.1.617.2 (Delta), displays modest cross-reactivity against SARS-CoV, and reduces viral replication in hamsters. The epitope coincides with a predicted "ideal" vulnerability site with high functional and structural constraints. Mutation of the residues of the conserved epitope variably affects FD20-binding but confers little or no resistance to neutralization. Finally, *in vitro* mode-of-action characterization and negative-stain electron microscopy suggest a neutralization mechanism by which FD20 destructs the Spike. Our results reveal a conserved vulnerability site in the SARS-CoV-2 Spike for the development of potential antiviral drugs.

**Keywords** COVID-19; cross-active neutralizing antibody; destruction of spike; receptor-binding domain; variants of concern
**Subject Categories** Microbiology, Virology & Host Pathogen Interaction; Structural Biology

## Introduction

The destructive spread of SARS-CoV-2 has caused a global public health crisis and economic and social disorder since its first emerge in December 2019. SARS-CoV-2 was initially expected to evolve slowly because of the proof-reading activity of its RNA polymerase, thus minimizing the chance for certain types of mutations. This genomic stability was thought to be good news for therapeutic design, but it became apparent in recent months, with the

1 State Key Laboratory of Molecular Biology, State Key Laboratory of Cell Biology, CAS Center for Excellence in Molecular Cell Science, Shanghai Institute of Biochemistry and Cell Biology, Chinese Academy of Sciences (CAS), Shanghai, China
2 University of CAS, Beijing, China
3 CAS Key Laboratory of Molecular Virology & Immunology, Institut Pasteur of Shanghai CAS, Shanghai, China
4 Hangzhou Center for Disease Control and Prevention, Hangzhou, China
5 Erasmus Laboratory Animal Science Center, Erasmus University Medical Center, Rotterdam, The Netherlands
6 European Research Infrastructure on Highly Pathogenic Agents (ERINHA-AISBL), Paris, France
7 Department of Viroscience, Erasmus University Medical Center, Rotterdam, The Netherlands
8 CAS Key Laboratory of Pathogenic Microbiology and Immunology, Institute of Microbiology, Center for Influenza Research and Early-warning (CASCIRE), CAS-TWAS Center of Excellence for Emerging Infectious Diseases (CEEID), CAS, Beijing, China
9 National Facility for Protein Science in Shanghai, Shanghai Advanced Research Institute (Zhangjiang Laboratory), CAS, Shanghai, China
10 Joint Center for Infection and Immunity, Guangzhou Institute of Pediatrics, Guangzhou Women and Children's Medical Center, Guangzhou Medical University, Guangzhou, China
11 Pasteurien College, Soochow University, Jiangsu, China
*Corresponding author. Tel: +86 571 85155039; E-mail: hzjkcjf@163.com
**Corresponding author. Tel: +86 21 54921620; E-mail: feilong.meng@sibcb.ac.cn
***Corresponding author. Tel: +86 21 54923190; E-mail: dlaville@ips.ac.cn
****Corresponding author (Lead contact). Tel: +86 21 54921434; E-mail: dianfan.li@sibcb.ac.cn
†These authors contributed equally to this work

emergence of B.1.1.7, B.1.351, B.1.617.2 (the Delta variant), and other variants, that SARS-CoV-2 is mutable and the mutants may compromise the effectiveness of the current neutralizing antibodies and vaccines (Collier *et al*, 2021; Wang *et al*, 2021).

The surface of SARS-CoV-2 is decorated by Spike (S), a trimeric glycoprotein that is encoded as a single polypeptide but is cleaved into two subunits S1 and S2 in the mature form. The S1 subunit contains an N-terminal domain (NTD) and a C-terminal receptor-binding domain (RBD) which recognizes the host receptor ACE2. RBD exists in two major conformations: the "up" conformation which is competent to engage ACE2, and the "down" conformation in which the ACE2-binding site, or so-called receptor-binding motif (RBM), is shielded by adjacent NTDs/RBDs (Hsieh *et al*, 2020; Wrapp *et al*, 2020). The RBD-ACE2 binding event triggers a conformational change that leads to membrane fusion and viral entry (Lan *et al*, 2020; Shang *et al*, 2020; Shang *et al*, 2020; Wang *et al*, 2020; Wrapp *et al*, 2020; Yan *et al*, 2020; Yuan *et al*, 2020). Because of this key role, the RBD has been an immunological "hot spot" for the development of neutralizing antibodies (Barnes *et al*, 2020; Cao *et al*, 2020; Hansen *et al*, 2020; Huo *et al*, 2020; Hurlburt *et al*, 2020; Ju *et al*, 2020; Liu *et al*, 2020; Pinto *et al*, 2020; Premkumar *et al*, 2020; Robbiani *et al*, 2020; Rogers *et al*, 2020; Shi *et al*, 2020; Wang *et al*, 2020; Wu *et al*, 2020b; Wu *et al*, 2020; Yuan *et al*, 2020; Zhou *et al*, 2020) and vaccines (Alsoussi *et al*, 2020; Dai *et al*, 2020; Tai *et al*, 2020; Walls *et al*, 2020; Yang *et al*, 2020), although neutralizing antibodies against the NTD and the S2 subunit have also been isolated (Chi *et al*, 2020; McCallum *et al*, 2021; Pinto *et al*, 2021).

The RBD structure features a core region consisting of 5 β-strands with connecting loops and α-helices, and the RBM which is mainly made of loops that lay on top of the core region (Lan *et al*, 2020). Current structurally characterized RBD-targeting monoclonal antibodies (mAbs) can be categorized into four classes based on the RBD conformation and their epitope positions relative to the RBM (Barnes *et al*, 2020). The Class 1 mAbs bind RBM in up-RBD; the Class 2 mAbs target RBM in both up- and down-RBDs; the Class 3 mAbs recognize non-RBM epitopes in up-RBD; and the Class 4 mAbs bind to non-RBM epitopes in both up- and down-RBDs. The RBM-targeting mAbs (classes 1 and 2) inhibit viral entry by directly competing with ACE2 (Barnes *et al*, 2020; Brouwer *et al*, 2020; Cao *et al*, 2020; Hansen *et al*, 2020; Liu *et al*, 2020; Shi *et al*, 2020; Wu *et al*, 2020). While effective against the wild-type virus, they rapidly lead to the generation of escape mutants *in vitro* and *in vivo* (Li *et al*, 2020; Starr *et al*, 2021; Wang *et al*, 2021). Examples include CB6 (also termed JS016 or LyCoV016) (Shi *et al*, 2020) which is in the steady progress of the clinical trial in China as single-antibody treatment, REGN-COV2 (REGN10933 + REGN10987) (Hansen *et al*, 2020) which are approved by the US Food and Drug Administration for emergency use authorization for treating COVID-19, and CV30 (Hurlburt *et al*, 2020), a human IgG isolated from infected COVID-19 patients. Classes 3 and 4 mAbs have some epitope residues in the core region (Huo *et al*, 2020; Pinto *et al*, 2020; Yuan *et al*, 2020; Zhou *et al*, 2020) which is more conserved (less mutable) (Greaney *et al*, 2020; Starr *et al*, 2020) than the RBM. Therefore, such mAbs are often cross-reactive (Huo *et al*, 2020; Pinto *et al*, 2020) and less susceptible to generation of neutralization escape mutants (Greaney *et al*, 2020) compared with RBM-targeting mAbs. However, non-RBM-targeting mAbs are less frequently reported. Well-

characterized members include the recently reported EY6A (Zhou *et al*, 2020), COVA1-16 (Lv *et al*, 2020), and two mAbs (CR3022 and S309) (Huo *et al*, 2020; Huo *et al*, 2020; Pinto *et al*, 2020; Yuan *et al*, 2020) that are originally developed against the closely related SARS-CoV which also caused a major outbreak in 2002 (Cherry & Krogstad, 2004). In responding to the aforementioned new variants, targeting different domains at the same time using antibody cocktails and developing a diversity of neutralizing mAbs targeting conserved domains are becoming critical.

Here, we report the isolation and characterization of a non-RBM-targeting mAb (FD20) from COVID-19 convalescent patients. FD20 binds the RBD with nanomolar affinity, neutralizes SARS-CoV-2 and several Variants of Concern with similar potency, and offers modest protection against viral replication and disease in hamsters challenged with live viruses. Structural characterization of FD20 reveals a conserved epitope and a neutralization mechanism by which FD20 destructs the S trimer. Mutagenesis of the epitope residues variably affects FD20-binding but has little or no effect on neutralization. Our results uncover a site of vulnerability in S for therapeutic development against SARS-CoV-2 and variants.

# Results

## Isolation and characterization of a neutralizing scFv with cross-reactivity

SARS-CoV-2 RBD-specific B cells were isolated from COVID-19 convalescent patients. A phage display library was constructed with mRNA from pooled RBD$^+$ B cells and used to select RBD-binders in the form of the single-chain variable fragment (scFv). ELISA-positive clones that caused earlier retention of the fluorescently labeled RBD on fluorescence-detection size exclusion chromatography (FSEC) were further screened using neutralizing assays with SARS-CoV-2 and SARS-CoV pseudovirus particles (pp) (Fig 1A). Seven neutralizing scFv were identified from 52 RBD binders using 50% neutralization at 1 μM or 25% neutralization at 0.4 μM as a cutoff (Appendix Fig S1, Appendix Table S1).

One of them, dubbed scFD20, bound RBD with a $K_D$ of 5.6 nM (Fig 1B) by biolayer interferometry (BLI) assay. Interestingly, scFD20 also bound with RBD from the closely related SARS-CoV (Fig 1C).

scFD20 neutralized SARS-CoV-2 pp (Fig 1D) with an $IC_{50}$ of 117.4 nM (3.8 μg/ml). Consistent with the cross-binding activity, scFD20 was cross-reactive against SARS-CoV pseudovirus, although with modest neutralizing activities (Fig 1D). To test the neutralizing activity in the bivalent form, we constructed the IgG format (termed FD20). The IgG form displayed higher neutralizing activity against both SARS-CoV-2 and SARS-CoV (Fig 1E), with an $IC_{50}$ of 12.0 nM (1.7 μg/ml) on SARS-CoV-2 and 15.1 nM for the D614G mutant.

## FD20 is resistant to several known escape mutants and variants of concern

The COVID-19 pandemic is evolving, with new lineages being reported all over the world. Among previous lineages, D614G has become globally dominant and N501Y has independently appeared in different 501Y-containing lineages (Leung *et al*, 2021). Of

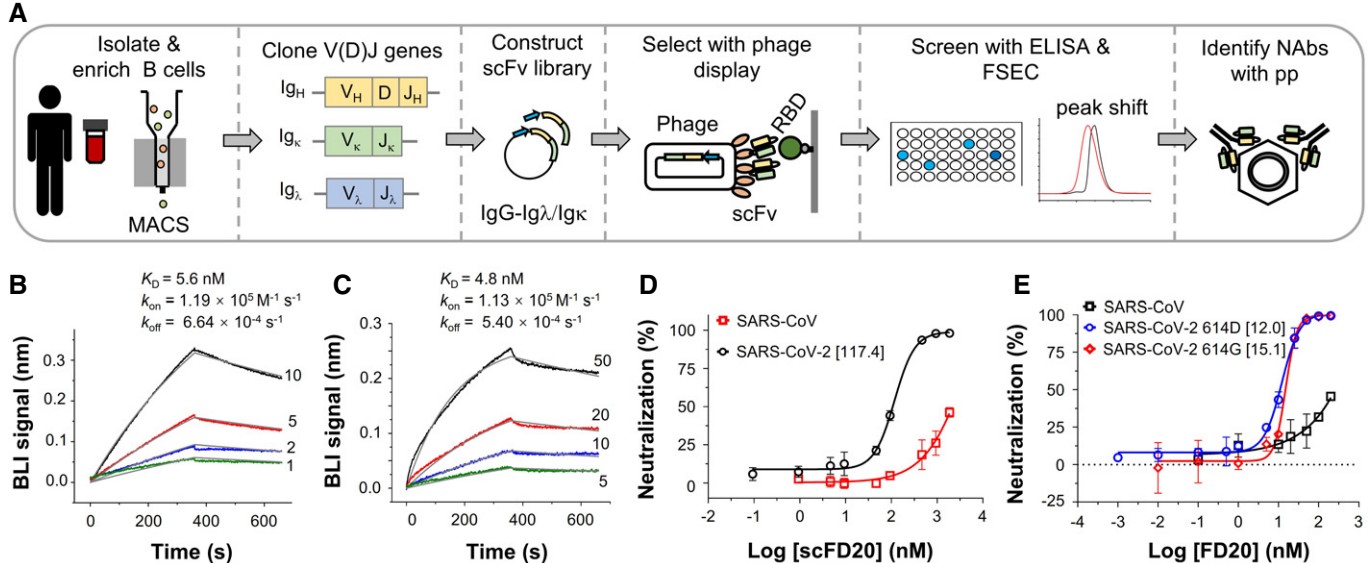

**Figure 1. Isolation and characterization of scFD20.**

A   Strategy for the screening of neutralizing antibodies. B cells from SARS-CoV-2 convalescent patients were enriched using magnetic-activated cell sorting (MACS) beads coated with biotinylated RBD. Phage display libraries expressing scFv were selected against RBD and the resulting clones were screened using ELISA, fluorescence-detection size exclusion chromatography (FSEC), and neutralization assays.

B, C   Kinetics of the binding between the scFv of FD20 (scFD20) and RBD from SARS-CoV-2 (B) or SARS-CoV (C). Biolayer interferometry (BLI) assay was performed with RBD immobilized and scFD20 as analyte at indicated concentrations (nM). Raw data are shown in color and fitted lines are shown in gray. $K_D$ values were calculated using 1:1 global fitting. Note, although the fitting of the binding curves in C gave a $K_D$ of 4.8 nM, the accuracy of the kinetic parameters may be compromised owing to the mismatch between the raw data and the fitted curves at the initial dissociation phase. The reason for the abnormal BIL profile is unknown but it may be caused by possible denaturation of the commercial SARS-CoV RBD during storage. It was also noted that the BLI signal in C was ~5 times less than that with SARS-CoV-2 RBD under similar conditions (B).

D   Neutralizing assay of scFD20 against SARS-CoV-2 (black) and SARS-CoV (red) pseudoviruses.

E   Neutralization assays using FD20 against pseudoviruses harboring the wild-type SARS-CoV-2 S (blue), S D614G mutant (red), or SARS-CoV S (black).

Data information: In D and E, numbers in brackets indicate $IC_{50}$ in nM calculated considering IgG as a dimer. Mean ± SD are plotted (*n* = 3 biological replicates). Source data are available online for this figure.

particular note, the recurrent emergence and transmission of deletion ΔH69/V70 often co-occur with the RBM mutants N501Y, N439K, and Y453F (Larsen *et al*, 2021; Leung *et al*, 2021; Oude Munnink *et al*, 2021; Tegally *et al*, 2021; Thomson *et al*, 2021). Four concerning lineages harboring most of these mutations are N501Y.V1 (B1.1.7, known as the Alpha variant; RBD mutation: N501Y) (Leung *et al*, 2021), N501Y.V2 (B.1.351, known as the Beta variant; RBD mutations: K417N, E484K, N501Y) (Tegally *et al*, 2021), P.1 (known as the Gamma variant; RBD mutations: K417T, E484K, and N501Y), and B.1.617.2 (known as the Delta variant; RBD mutations: K417N, L452R, T487K) which emerged in the UK, South Africa, Brazil, and India, respectively. Several naturally occurring RBD mutations were shown to abrogate interactions with known mAbs and to reduce immune sera binding, raising concerns that more escape mutants could emerge (Li *et al*, 2020; Starr *et al*, 2021; Tegally *et al*, 2021; Wang *et al*, 2021; Yao *et al*, 2021). Notably, FD20 displayed similar neutralizing activity against SARS-CoV-2 pp harboring S mutants from B.1.1.7/P.1/B.1.617.2 and slightly reduced activity against B.1.351 ($IC_{50}$ value increased to ~2 fold) (Fig 2A).

Using replication-competent live virus, we compared the neutralization of the initial Wuhan isolate (hCoV-19/China/CAS-B001/2020), an early isolate in Germany (betaCoV/Munich/BavPat1/2020), and the UK/B.1.1.7 variant. Similar to the results with pseudovirus, FD20 showed no dramatic differences in neutralizing

activity against the three isolates (hCoV-19/China/CAS-B001/2020, 5.2 nM; betaCoV/Munich/BavPat1/2020, 11.8 nM; UK/B.1.1.7, 7.9 nM) (Fig 2B and C).

The cross-reactivity (Fig 1E) and the ability to neutralize the four Variants of Concern suggest that FD20 is more resistant than RBM-targeting mAbs to SARS-CoV-2 polymorphism. To test this, we used pseudotyped virus entry assays with selected S escape mutants or variants (Fig 2D) that bear several mutations in the recent literature (Larsen *et al*, 2021; Leung *et al*, 2021; Oude Munnink *et al*, 2021; Tegally *et al*, 2021; Thomson *et al*, 2021). Contrary to RBM-targeting mAbs (CB6, CV30, REGN10933) for which the $IC_{50}$ were affected variably by 50–2,000 times (Fig 2D), and REGN10987 (a neutralizing mAb mostly by steric hindrance) for which the $IC_{50}$ was affected by E406W (~80 fold) and N439K/D614G (~30 fold), FD20 did not show noticeable susceptibility to the SARS-CoV-2 S mutations. In particular, E406W (Starr *et al*, 2021) and K417N (Li *et al*, 2020; Starr *et al*, 2021), two mutants that attenuate the Regeneron cocktail and CB6, respectively, did not show resistance to FD20.

**Evidence for SARS-CoV-2 neutralization *in vivo***

We next investigated the *in vivo* potential of FD20 by intraperitoneal injection 6 h before intranasal inoculation of hamsters by live viruses, followed by assessing the effect of FD20 treatment on body

weight (Fig 3A), lung titer (Fig 3B and C), nasal titer (Fig 3D and E), and pathology in the lung (Fig 3F and G). At 2 days post-inoculation (d.p.i.), infected hamsters displayed reduced activity and began to progressively lose weight throughout the study (up to ~15% reduction in untreated animals). There was a significant reduction in weight loss associated with FD20 treatment compared to PBS at 4 d.p.i. (Fig 3A), as well as a small but significant reduction in viral RNA in the lungs of hamsters treated with FD20 (5 mg doses; 62–75 mg/kg) compared to those that were untreated ($P = 0.005$) (Fig 3B). This reduction in lungs correlates with the quantification of the positive area to SARS-CoV2 antigen (IHC) for which FD20-treated hamsters have a strong reduction (~3 fold) in staining compared to untreated hamsters (Fig 3G).

When assessed using $TCID_{50}$, no significant changes were observed between the FD20 group and the PBS group (Fig 3C), due

to animal-to-animal variation, although 2 out of 5 animals showed no evidence of infectious viruses in the lungs (Fig 3C) or nose (Fig 3 E) after treatment. No significant reduction of viral RNA was observed in the nose (Fig 3D) and the quantification of the bronchiolar or parenchymal inflammation area pathology using hematoxylin and eosin (H&E) did not show significant improvement upon FD20 treatment (Fig 3F). Overall, the results show modest neutralizing activity of FD20 *in vivo*. Nevertheless, the fact that FD20 is broadly active warrants further optimization such as *in vitro* maturation to improve potency.

## FD20 engages RBD at a surface distal to RBM

To characterize FD20-RBD interactions, we solved the crystal structure of the scFD20-RBD complex at 3.1 Å (Fig 4A and Table 1). The

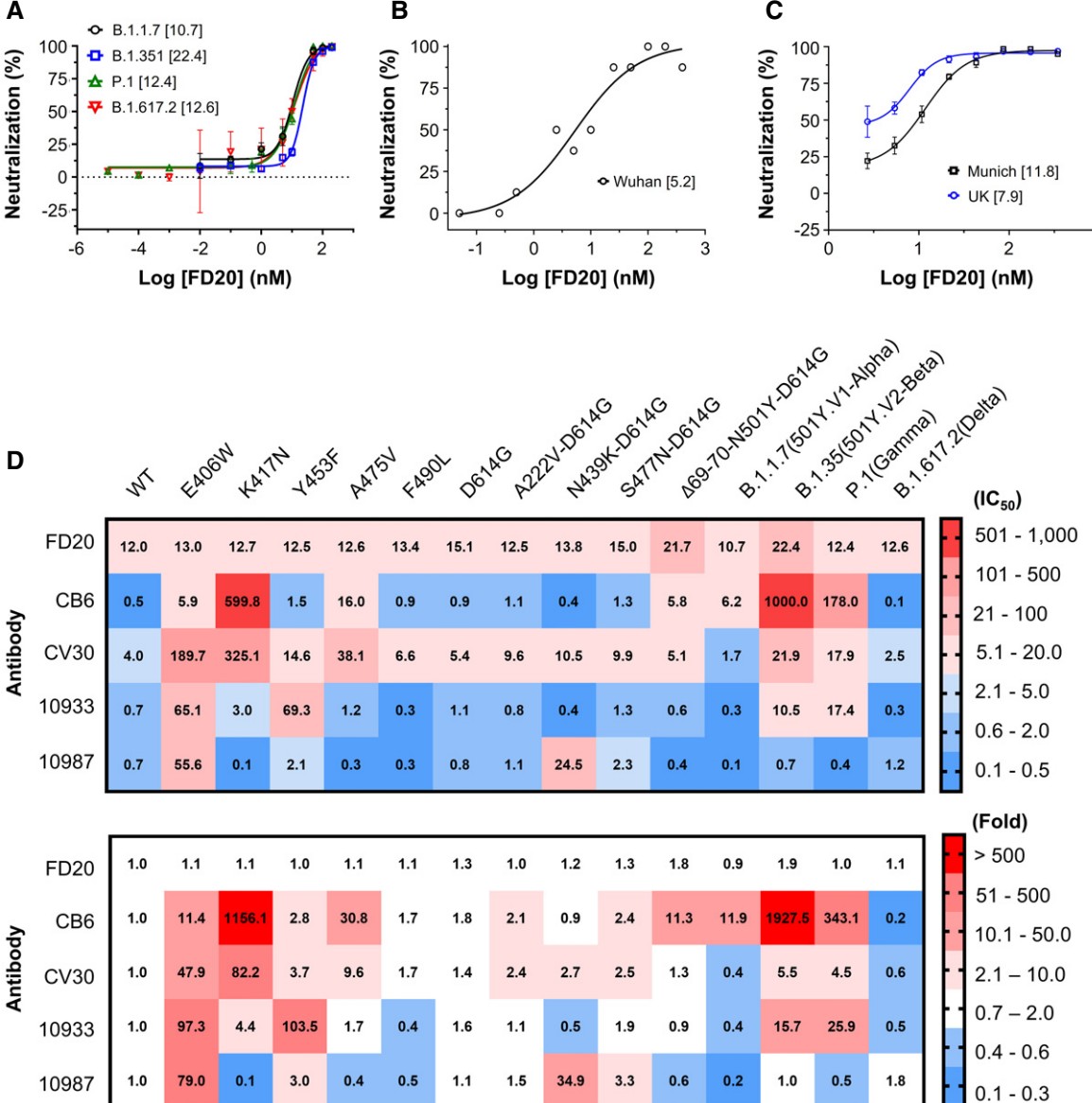

**Figure 2.**

◄

**Figure 2. FD20 is resistant to several escape SARS-CoV-2 S mutants.**

A Neutralization assay of FD20 using SARS-CoV-2 pp harboring spike derived from four Variants of Concern as indicated. Mean ± SD are plotted ($n = 3$ biological replicates). Numbers in brackets indicate $IC_{50}$ values in nM.

B Microneutralization assay of FD20 against authentic SARS-CoV-2 Wuhan isolate (hCoV-19/China/CAS-B001/2020). The protective effect of antibody was assessed using cells infected with a high viral dose (2,000 $TCID_{50}$ ml$^{-1}$). Data are from a single experiment performed in octuplicates. Numbers in brackets indicate $IC_{50}$ values in nM.

C The plaque-reduction neutralization assay using two authentic SARS-CoV-2 strains: the Germany isolate (BetaCoV/Munich/BavPat1/2020) and UK/B1.1.7. Mean ± SD are plotted ($n = 3$ biological replicates). Numbers in brackets indicate $IC_{50}$ values in nM.

D $IC_{50}$ (top, in nM) and normalized $IC_{50}$ folds (bottom, normalized against the wild-type) of FD20 and control antibodies measured using different SARS-CoV-2 pp harboring S mutants or variants. E406W is an escape mutant from REG10933 and REG10987 cocktail generated *in vitro* (Starr *et al*, 2021). K417N is an escape mutant for CB6 (LyCoV.016) and is present in the B.1.351 lineage first identified in South Africa (Li *et al*, 2020; Starr *et al*, 2021). Y453F increases the ACE2 binding affinity, and with A475V and F490L, is found in independent mink-related SARS-CoV-2 variants (Larsen *et al*, 2021; Oude Munnink *et al*, 2021) identified in Denmark and the Netherlands (B1.1 and B.1.1.298) that may escape human antisera and REG10933 (Starr *et al*, 2021). D614G appeared early during the pandemic and is now the dominant form worldwide and is associated with other mutations in new variants of concern (Korber *et al*, 2020). A222V and S477N are described in 20A.EU1 (B.1.177) and 20A.EU2 (B.1.160) that emerged in early summer 2020 and subsequently spread to multiple locations in Europe at the end of 2020 (Lemey *et al*, 2021). N439K is in B.1.141, B.1.258, and in mink strains; it increases affinity to ACE2 and reduces neutralization of sera from convalescent patients (Thomson *et al*, 2021). Finally, Δ69/70-N501Y-D614G are characteristic mutations in RBD for the UK/B.1.1.7 lineage (501Y.V1) (Leung *et al*, 2021), one of the three fast-spreading new variants of SARS-CoV-2 that have emerged in recent months. The B1.351 (501Y.V2) identified in South Africa has raised concerns that the efficacy of current vaccines and therapeutic monoclonal antibodies could be threatened (Tegally *et al*, 2021). The P.1 (Gamma or Brazilian variant) has contributed to a surge in cases in the northern city of Manaus and it has been reported in January 2021 in Japan. The highly contagious B.1.617.2, commonly known as the Delta, is currently responsible for over 90% of the cases in the UK and over 80% of the cases in the United States and has caused concerns about vaccine efficacy (Lopez Bernal *et al*, 2021). The background is color-colored based on the neutralizing activity (top) or change in neutralizing activity (bottom). $IC_{50}$ values are the mean from at least two independent experiments.

Source data are available online for this figure.

structure was refined to $R_{work}/R_{free}$ of 0.2518/0.2763. The asymmetric unit contained one RBD and one single-chain variable fragment scFD20.

The RBD structure resembles a high chair with a short backrest and scFD20 binds almost directly under the "backrest" region (Fig 4 A) with a surface area (Krissinel & Henrick, 2007) of 629.64 Å$^2$. Among the six total complementarity-determining regions (CDRs), three are directly involved in RBD binding. They include the heavy chain *H*-CDR3 and light-chain *L*-CDR1 and *L*-CDR2 (Fig 4B). Despite a modestly sized surface, scFD20 forms a rich network of interaction with the RBD, including 4 salt-bridges, 10 hydrogen bonds, and hydrophobic interactions mostly contributed by three tyrosine residues (Tyr108, Tyr110, and Try112) which is part of a 5-tyrosine cluster in the *H*-CDR3 (Appendix Fig S1B).

Structural alignment of scFD20-RBD to the ACE2-RBD structure (Lan *et al*, 2020) showed that the epitope of FD20 is distant from the ACE2 (receptor)-binding motif (RBM) (Fig 4C). Nevertheless, the binding of FD20 inhibited ACE2-RBD interactions (Fig 4D). Probably, slight distortions at RBM caused by FD20 binding (AppendixFig S2) were enough to perturb ACE2-RBD binding because interactions at the non-linear RBM should depend on the proper formation of a precise three-dimensional shape. Such allosteric inhibition (Huo, Zhao, *et al*, 2020; Zhou *et al*, 2020) is also observed in the case of CR3022 and EY6A.

**Mutation of the conserved epitope has little effect on FD20's neutralizing activity**

In accord with FD20's broad neutralizing activity, sequence and structural alignment of SARS-CoV and SARS-CoV-2 at the RBD region showed high conservation of the FD20 epitope (Fig 4E, Appendix Fig S3A and B). Among the twelve residues at the contact site, nine are identical, two are highly conservative changes (Arg/Lys swapping), and one is a modestly conservative change (Asn to Glu) (Fig 4E). The great similarity between the two RBDs is consistent with FD20's cross-reactivity in both binding and neutralization to SARS-CoV (Fig 1C–E). Complete conservation of the epitope residue was observed when comparing to RaTG13 (Ge *et al*, 2016), of which the RBD is 89.5% identical and 94.0% similar to SARS-CoV-2 RBD (Fig 4E) (note, residues Pro330 and Lys529 of SARS-CoV-2 S was used to calculate sequence identity/similarity). Among five other coronaviruses known to infect humans, three of them (hCoV-OC43, hCoV-HKU, and MERS) contain RBD-like structures (Ou *et al*, 2017; Yuan *et al*, 2017; Tortorici *et al*, 2019). Although the sequence homology at the FD20's epitope was only modest due to their overall low sequence similarity in the RBD (Appendix Fig S3C), the 3-dimensional organization of the epitope region is structurally conserved (Fig 4F).

In addition to the conservation, the FD20 epitope residues are reported to be functionally or structurally constrained. In a recent report, Starr *et al* (2020) predicted antibody epitopes on RBD based on the tolerance to mutation regarding expression (structural) and ACE2-binding affinity (functional) using a deep mutational scanning approach. The predicted ideal epitope (Fig 4E and G), surrounding Glu465, consists of 10 residues, 8 of which are surprisingly contained in the FD20 epitope (12 residues) (Fig 4E). The two exceptions, Arg454 and Asp467, are also in proximity (Fig 4G). Because functional and structural constraints are known to correlate with conservation, the analysis reinforces the idea that the FD20 epitope is less likely to mutate. Indeed, the mutation frequencies of the FD20 epitope residues from ~1.5 million deposited sequences (Elbe & Buckland-Merrett, 2017) (cov.lanl.gov) were relatively low. Thus, there is only one mutation reported for Arg355 (R355T) (as of June 9, 2021), and the mutation frequencies (per million) of the rest residues were below 70, with the exception for Asn354 which was 391 (Fig EV1).

To further test whether the naturally occurring mutations affect FD20's binding and neutralizing activity, we have designed a panel of mutants as listed in Table 2 using the following criteria. For each epitope residue, the most frequent mutation was chosen unless the substitution was conservative, in which case amino acid with the most dramatic changes was selected. For example, Arg466 was

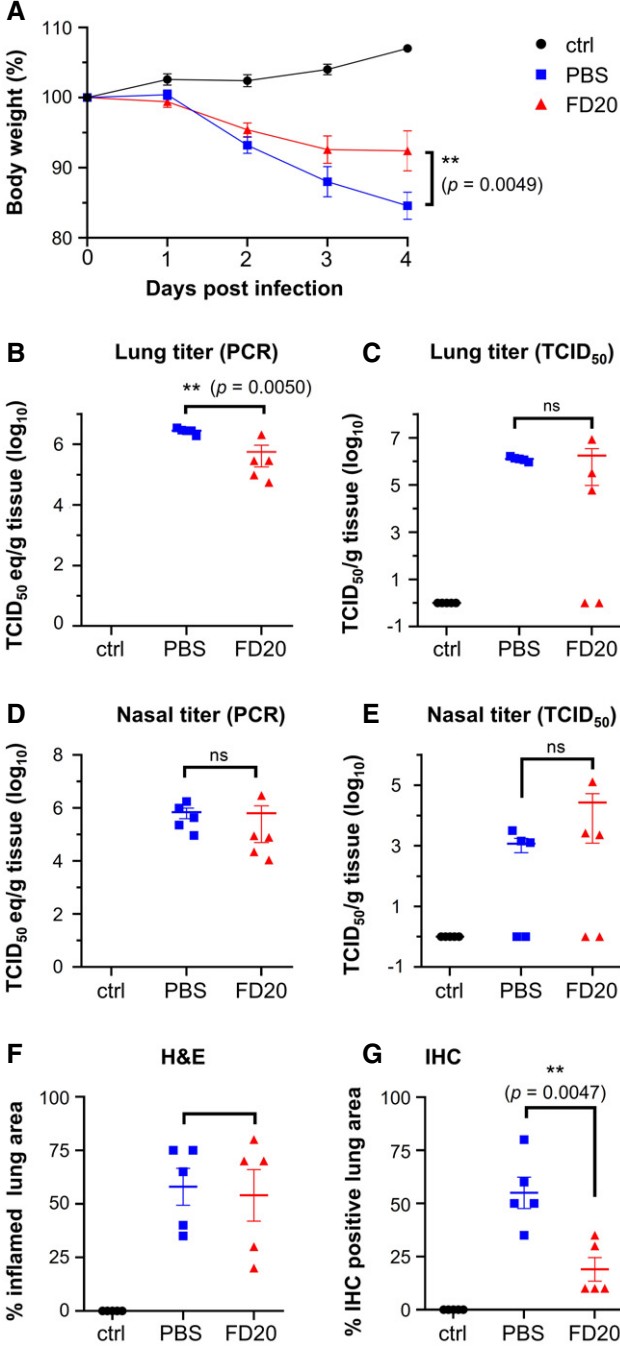

Figure 3.  FD20 reduces replication and pathology in the susceptible hamster model.

Hamsters were inoculated intranasally with $10^5$ $TCID_{50}$ of SARS-CoV-2 (isolate BetaCoV/Munich/BavPat1/2020) and treated with FD20 (red) or PBS buffer (blue). Uninfected hamsters were used as controls (ctrl, black). None of the hamsters in the study died or met euthanasia criteria before study termination at 4 d.p.i.

A   Body weights of hamsters treated with antibodies were measured at indicated days after inoculation with SARS-CoV-2. The mean % of starting weight $\pm$ SEM ($n = 5$ biological replicates) are plotted.

B–E   SARS-CoV-2 viral RNA (B and D) or infectious virus (C and E) was detected in the lung (B and C) and nasal turbinates (D and E). The mean copy number or the mean infectious titer is shown, error bars represent SEM ($n = 5$ biological replicates).

F, G   Percentage of inflamed lung tissue (F) and percentage of lung tissue expressing SARS-CoV-2 antigen (G) estimated by microscopic examination in different groups of hamsters at 4 days after SARS-CoV-2 inoculation. Individual (symbols) and mean (horizontal lines) percentages are shown. Error bars represent SEM ($n = 5$ biological replicates).

Data information: In A, Statistics were performed using two-way ANOVA followed by Sidak's multiple comparisons test. Time point starting to show significance: 5 d.p.i. In B-G, statistical analyses were performed using 2-tailed, unpaired Student's *t*-test on log-transformed data (B–E) or raw data (F, G). ns, not significant. Data for the two controls (PBS and control) were replotted from ref. (Li *et al*, 2021) with permissions.
Source data are available online for this figure.

infectivity (Fig EV2C) and neutralization assay (Fig EV2D). In addition, the expression level and processing of S (whether S is cleaved to produce S1 and S2) were assessed by Western blotting as quality control for trafficking and maturation (Fig EV2E and F).

Among the 15 tested RBD mutants, W353R, R355T, and D467Y showed no expression in insect cells (Table 2). Although the corresponding full-length S mutants were detected in mammal cells, they were not processed effectively as evidenced by the lack of the S1 subunit on SDS–PAGE (Fig EV2E and F). These results jointly suggest the mutants were misfolded. As expected, mutations of Lys462 and Glu465 attenuated FD20-binding. However, the mutations either reduced infectivity drastically (E465K, 5 % of the wild-type), or remained sensitive to FD20, displaying similar $IC_{50}$ values (Fig EV2D, Table 2). Likewise, mutation of Arg466, another residue that contributes a salt bridge, also impaired binding but had little effect on the neutralization (Table 2). Finally, the rest of the mutation did not weaken the binding activity, and the neutralizing activities of FD20 for these mutants also did not change noticeably (Table 2). The two most FD20-resisting mutants were D428G and K462A, which showed a $\sim$ 2-fold $IC_{50}$ compared to the wild-type SARS-CoV-2 pp. Taken together, the analyses showed that the FD20 epitope is highly conserved, and the experimental results confirmed that some (Trp353/Asp467/Arg355/Glu465) are indeed highly structurally constrained as suggested by Starr *et al* (2020); in the case of mutants with similar infectivity to the wild-type, FD20 remained effective for neutralization, indicating FD20's potential to resist escape mutants.

## Compared with mAbs at (pre)-clinical stages, FD20 strongly inhibits cell–cell fusion

Besides fusion mediated by virions, S proteins present at the plasma membrane can trigger receptor-dependent cell–cell fusion, which leads to the formation of giant multinucleated cells (syncytia), a

mutated to isoleucine instead of lysine. In addition, the charge of Lys462 and Glu465 were either eliminated by alanine mutation (not naturally occurring) or reversed by the mutations K462E and E465K (found in the database with low frequencies) (Fig EV1B), respectively; such mutations were expected to weaken the FD20-RBD binding because both residues provided intermolecular salt bridges (Fig 4B). Finally, the epitope-proximal residue Asp467 was mutated to tyrosine to test its structural/functional constraint. The mutations were introduced to RBD for binding assay separately with ACE2 (Fig EV2A) and FD20 (Fig EV2B), and to S protein in SARS-CoV-2 pp for

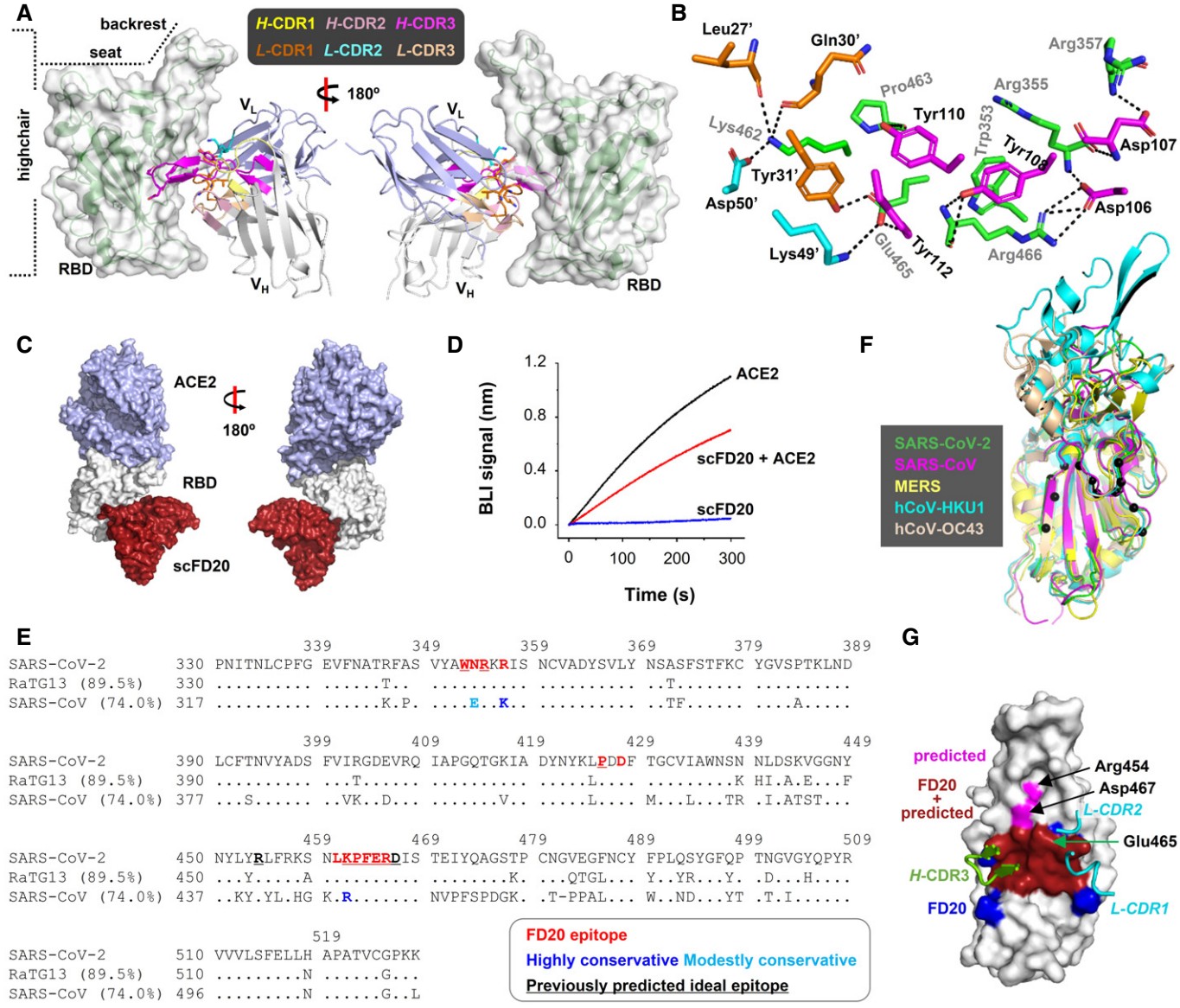

**Figure 4.  FD20 binds RBD at a conserved region previously predicted as an "ideal" epitope with high functional and structural constraints.**

A   The overall structure of the scFv FD20 (cartoon) in complex with SARS-CoV-2 RBD (green cartoon in semi-transparent surface). $V_H$ and $V_L$ denote heavy and light chain variable domains, respectively. The $V_H$ CDRs and $V_L$ CDRs are colored as indicated. The RBD structure resembles a highchair that has a "seat" region and a short "backrest" region.

B   Detailed interactions between residues from the RBD (green) and scFD20 involve $V_H$ CDR3 (magenta), $V_L$ CDR1 (orange), and $V_L$ CDR2 (cyan). FD20 residues are labeled black and RBD residues are labeled gray. A prime symbol indicates residues from $V_L$ CDRs.

C   scFD20 (dark red) binds the RBD (white) at a site distal to that of ACE2 (light blue).

D   Modest perturbation of scFD20 for RBD-ACE2 interaction. A sensor with RBD immobilized was soaked in 200 nM of scFD20 before being further soaked in scFD20-containing buffer with (red) or without (blue) 25 nM of ACE2 for BLI signal recording. As a control, the ACE2-RBD interaction was monitored in the absence of scFD20 (black).

E   The FD20 epitope (red) is conserved between SARS-CoV-2, RaTG13, and SARS-CoV. Dots indicate identical residues and dashes indicate gaps. Brackets indicate sequence identity between the aligned RBD and SARS-CoV-2 RBD. NCBI accession codes for the sequences are as follows: RaTG13, QHR63300.2; SARS-CoV, NC_004718.3.

F   The FD20 epitope (Cα in black sphere) is similarly organized in three-dimension among the coronaviruses that contain an RBD-equivalent region. The PDB accession codes are: SARS-CoV-2, 7CYV (this work); SARS-CoV, 2G75 (Prabakaran *et al*, 2006); MERS, 5X59 (Yuan *et al*, 2017); hCoV-HKU1, 5GNB (Ou *et al*, 2017); and hCoV-OC43, 6OHW (Tortorici *et al*, 2019).

G   The predicted "ideal" epitope (red + magenta) is mostly contained in the FD20 binding site (red + blue). The three CDRs (green for heavy chain and cyan for light chain) that are in close contact with RBD (white) are shown. A green arrow indicates Glu465 which is at the center of the predicted "ideal" epitope (Starr *et al*, 2020).

**Table 1. Statistics for data collection and refinement of scFD20-RBD.**

| | FD20-RBD |
|---|---|
| Data collection | |
| Space group | C 1 2 1 |
| Cell dimensions | |
| a, b, c (Å) | 206.96 57.93 47.21 |
| α,β, γ (°) | 90.0 100.43 90.0 |
| Wavelength (Å) | 0.97915 |
| Resolution (Å) | 46.43–3.13 (3.24–3.13)[a] |
| $R_{merge}$ | 0.2724 (1.125) |
| $R_{pim}$ | 0.1322 (0.5387) |
| $I/\sigma(I)$ | 5.65 (1.46) |
| Completeness (%) | 98.89 (94.68) |
| Multiplicity | 5.2 (5.3) |
| $CC^{*b}$ | 0.995 (0.94) |
| Refinement | |
| Resolution (Å) | 46.43 – 3.13 |
| No. reflections | 9,764 |
| $R_{work}/R_{free}$ | 0.2518/0.2763 |
| No. atoms | 3,225 |
| Protein | 3,166 |
| Ligand/ion | 59 |
| Water | 0 |
| No. residues | 429 |
| B-factors (Å$^2$) | 61.91 |
| Protein | 61.49 |
| Ligand/ion | 84.55 |
| R.m.s deviations | |
| Bond lengths (Å) | 0.004 |
| Bond angles (°) | 0.800 |
| Ramachandran | |
| Favored (%) | 95.95 |
| Allowed (%) | 4.05 |
| Outlier (%)a | 0 |
| PDB ID | 7CYV |

[a]Highest resolution shell is shown in parenthesis.

[b]$CC^* = \sqrt{\frac{2CC_{1/2}}{1+CC_{1/2}}}$.

common phenomenon observed post-mortem in lung tissues (Tian et al, 2020). To characterize FD20's mode of action, we first evaluated its capacity to inhibit cell–cell fusion compared to other antibodies in clinical trials or with emergency use authorization for treating COVID-19. They include REGN10933, REGN10987, CB6, and CV30. In addition, the cross-active CR3022 was also included. Their $IC_{50}$ values, measured on retroviral pseudotypes harboring the wild-type or D614G Spike, were similar to the data from the literature (Fig EV3A and B) (Hurlburt et al, 2020; Shi et al, 2020). In parallel, we produced an FD20 mutant (Y112R) that compromised RBD binding as a control.

Cell–cell fusion inhibition was measured using a luciferase reporter that is activated in trans after mixing of cytoplasmic contents. Surprisingly, although FD20 was not the best antibody in the entry assay (Fig EV3A and B) having an $IC_{50}$ higher by 3- to 22-fold compared to other mAbs, it was the most potent antibody to inhibit cell–cell fusion (Fig 5A and B). Indeed, at 1 μM, FD20 inhibits fusion at 90%, whereas the REG10933 and REG10987 inhibit fusion at 60% maximum. Even at a 10 nM concentration, the cell–cell fdusion was effectively suppressed by FD20, displaying a ∼ 60% inhibition. This feature may be of therapeutic interest to prevent syncytia formation in vivo.

**Unlike RBM-targeting mAbs, FD20 inhibits both *binding* and *post-binding* steps of viral entry**

Virus entry is initiated by the attachment to receptors and is followed by conformational changes of viral proteins, which leads to the fusion of virus and cellular membranes. During the entry process, inhibitors can act at a *pre-binding* step (a virucide effect that impairs particle), during *binding* (by competitive inhibition of ACE2 binding), or at a *post-binding* step (inhibition of the conformational changes that leads to membrane fusion) (Appendix Fig S4). When the mAbs were added after the *binding* step, an inhibition of entry, although modest in virological context (25%), was detected for FD20 but not for RBM-targeting mAbs (Figs 5B and EV3C), suggesting that FD20 acts at a *post-binding* step. As a control, pre-incubation of FD20 with cells (Fig EV3D, condition "2") did not inhibit infection, ruling out the possibility of nonspecific effects on cells and thus confirming that the FD20's *post-binding* effect is on SARS-CoV-2 particles.

To investigate if FD20 also acts on particles at the *pre-binding* step, we pre-incubated, or not, the SARS-CoV-2 pp with FD20 before infection (Fig 5C). The pre-incubation increased the neutralizing activity of FD20 by ∼32% (Figs 5C and EV3D). By contrast, the increase was less for all the control mAbs (Figs 5C and EV3D and E). The mechanism for the stronger pre-incubation effect of FD20 will be investigated further below and the weak pre-incubation effect by the control mAbs was probably due to residual antibodies that remained bound with pp.

To assess the kinetic consequences of the *pre-binding* effect, we developed a binding assay using virus-like particles (VLP). We produced VLP-GFP by expressing S, N, E, and GFP-M (fusion protein of green fluorescent protein to M). The purified VLP can bind VeroE6-hACE2 cells and the binding specificity was confirmed using soluble ACE2 as an inhibitor (Fig 5D). Using this assay, we demonstrated that pre-incubating of FD20 with the VLP inhibited the attachment of the particles to cells, similarly to the control antibodies.

**FD20 binding irreversibly impairs virus particle's infectivity**

The efficient *pre-binding* impact of FD20 suggests a possible irreversible virucide effect. To test this, we pre-incubated SARS-CoV-2 pp with FD20 and diluting the mix before infection (Group *5*) (Fig 6A and B) to reach an FD20 concentration below its efficient neutralizing activity (determined by Group *3*) (Fig EV3A). Antibodies acting on virus reversibly (such as binding and dissociation) are expected to lose neutralizing activity when diluted (which is

**Table 2. Summary of biophysical and biological characteristics of mutants at the FD20 epitope.**

| Constructs[a] | Mutation frequency (per million)[b] | ACE2-binding[c] | FD20-Binding[c] | S Expression[e] | S Processing[f] | Infectivity (% of wt) | IC50, nM (fold) |
|---|---|---|---|---|---|---|---|
| Wild-type | n/a | + + | + + | + + | + + | 100 | 12.0 (1.0) |
| W353R | 2.7 (4.1) | n.d.[d] | n.d.[d] | + + | − | 0.7 | n.d.[d] |
| N354K | 172.0 (391.7) | + + + | + + | + + | + + / − | 85.6 | 11.9 (1.0) |
| R355T | 0.7 (0.7) | n.d.[d] | n.d.[d] | + + | − | 0.6 | n.d.[d] |
| P426S | 14.3 (16.3) | + + + | + + | + + | + | 55.7 | 12.7 (1.1) |
| D428G | 15.0 (29.3) | + + | + + | + + | + + / − | 72.1 | 23.7 (2.0) |
| L461I | 8.8 (10.2) | + + + | + + | + + | + + | 92.6 | 14.8 (1.2) |
| K462A | 0 (21.1) | + + | + | + + / − | + + | 119.0 | 25.2 (2.1) |
| K462E | 5.1 (21.1) | + + + | + / − | + + | + + | 67.9 | 16.6 (1.4) |
| P463S | 57.2 (62.0) | + + | + + | + | + / − | 40.5 | 12.4 (1.0) |
| F464S | 1.4 (9.5) | + + + | + + | + | + / − | 26.0 | 15.9 (1.3) |
| E465A | 0 (21.1) | + + + | + / − | + | + + / − | 38.3 | 16.7 (1.4) |
| E465K | 0.7 (21.1) | + + + | + / − | + | − | 5.0 | n.d.[d] |
| R466A | 0 (10.2) | + + / − | + + / − | + | + / − | 21.4 | 17.7 (1.5) |
| R466I | 2.7 (10.2) | + + | + + / − | + | + | 77.1 | 15.0 (1.3) |
| D467Y | 1.4 (4.1) | n.d.[d] | n.d.[d] | + + | − | 0.8 | n.d.[d] |

[a]Binding was performed with the wild-type or mutations on RBD, and infectivity and neutralization assays were performed with pseudoviruses harboring the wild-type or mutant S protein.

[b]Frequency of the particular mutation followed by mutation frequency at this residue in brackets as of June 9, 2021 (www.gisaid.org; cov.lanl.gov). "0", no such mutation has been reported.

[c]Binding properties were evaluated by the biolayer interferometry (BLI) signal and apparent binding affinity.

[d]Not determined because either the construct had no expression in insect cells (for binding) or the mutant SARS-CoV-2 pp had low infectivity (for neutralizing assay).

[e]Expression level of S was evaluated by Western blotting using an anti-S1 antibody.

[f] Proper processing and trafficking were evaluated by Western blot of S1 in the cell culture supernatant. Binding kinetics and expression level were rated from high to low following the sequence of "++", "++/−", "+", "+/−", and "−". The data for binding (BLI curve), expression (Western blot), infectivity, and neutralization are in Fig EV2. Source data for infectivity and IC50 experiments are available online.

characterized by a reduction in normalized neutralizing activity from Group 4 to Group 5). By contrast, mAbs with irreversible effects (such as ultra-tight binding and S destruction) should display similar normalized neutralizing activity between Group 4 and Group 5.

As shown in Fig 6B, compared to the undiluted condition (Group 4, Fig 6A), the dilution reduced the neutralizing activity of FD20 (Group 3), but not when a pre-incubation is included (Group 5). By contrast, the pre-incubation did not prevent loss of neutralizing activity for all other mAbs (orange bars, Fig 6B), suggesting that FD20, unlike other mAbs, is an irreversible inhibitor. This irreversible effect remained when the assay was carried out in the presence of the control mAbs (green bars), suggesting that FD20 can act on SARS-CoV-2 independently, consistent with the structural observation that FD20's epitope does not share with the control mAbs.

To further investigate whether FD20 can inhibit SARS-CoV-2 infection synergistically with the control mAbs, we performed neutralization experiments using the mAb cocktails; the non-neutralizing FD20(Y112R) mutant was included as a control. The combination of FD20 and CV30 showed a mild synergistic effect, as evidenced by the slightly lower IC50 value for the mix than that for either mAbs alone (Fig EV3F), while the IC50 values remained unchanged for both FD20 and RBM-targeting mAbs for the rest of them. It may be because the IC50 of CV30 is the closest to FD20

(12.0 nM versus 4.0 nM), whereas the IC50 values for CB6 (0.5 nM), REG10933 (0.7 nM), and REG10987 (0.8 nM) are much lower than FD20 and therefore masked the potential improvement by FD20 addition.

**FD20 neutralizes SARS-CoV-2 by destructing Spike**

Aligning the FD20-RBD structure to the full-length Spike structure showed that the FD20 epitope is inaccessible. As shown in Fig EV4A, all three FD20-binding sites are buried between the RBD and the N-terminal domain of S1 (NTD) of the adjacent monomers in the "closed" state (Walls et al, 2020) when the three RBDs are at a "down" conformation. When aligned to the "open" state with one "up"-RBD, the epitope is more exposed (Fig EV4B). Nevertheless, notable clashes are still observed between FD20 and NTD including the two glycans linking to Asn165 and Asn234 (Fig EV4C) (Walls et al, 2020). This raises the possibility that FD20 executes its virucide effect by destructing S, as reported for CR3022 and EY6A (Huo et al, 2020; Zhou et al, 2020).

To test this hypothesis, we performed negative-stain electron microscopy of S-2P (an engineered S mutant containing two stabilizing proline mutations, see Materials and Methods) to investigate the impact of FD20 on the S-2P's integrity. The purified S-2P, but not FD20, showed particles with its typical "chicken leg" shape (Fig 6C

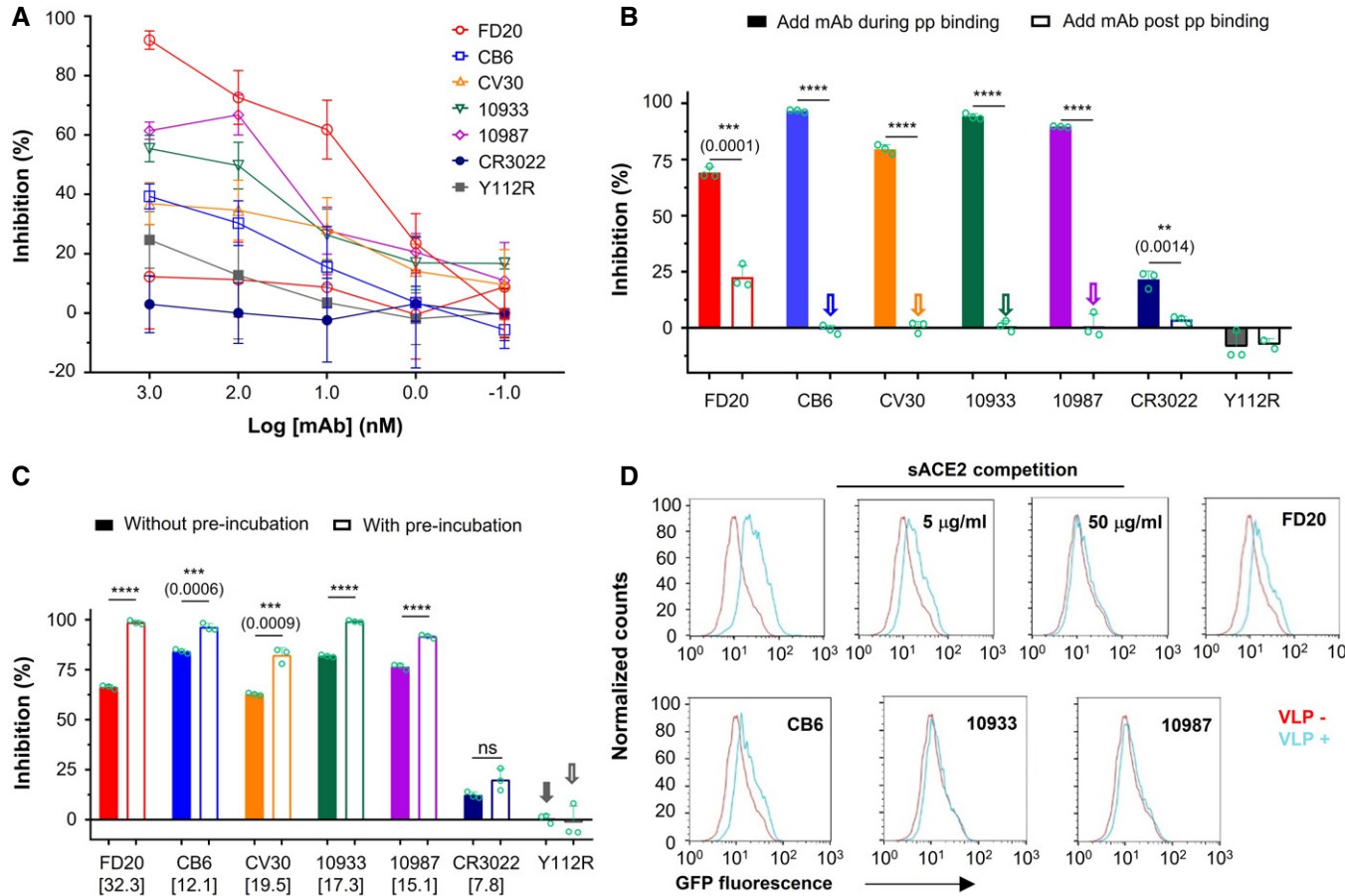

**Figure 5. FD20 acts on Spike to inhibit cell–cell fusion and infection.**

A   Inhibition of cell–cell fusion by mAbs. Cell fusion was quantified by measuring the luciferase expression induced by full cell membrane fusion and mixing of cytoplasm, and the rate of inhibition by indicated antibodies compared to the condition without antibody is reported. Data depict mean ± SD (*n* = 4 biological replicates).

B   Inhibition of a *post-binding* step by FD20, but not by other mAbs. Antibodies were added to VeroE6-hACE2 cells during (filled) or after (open), the *binding* step of SARS-CoV-2 pp. Mean ± SD are plotted (*n* = 3 biological replicates). As control, cells were incubated at each step with an equal volume of PBS or Y112R. *P* < 0.0001 (****) unless specified otherwise (2-tailed, unpaired Student's *t*-test).

C   FD20 pre-incubation with SARS-CoV-2 pp in the absence of cells strongly impairs entry. To test direct action on particles, SARS-CoV-2 pp and FD20 were co-incubated for 1 h, and then, the premixes were used to infect VeroE6-hACE2 cells for 6 h at 37°C. Mean ± SD are plotted (*n* = 3 biological replicates). Numbers in brackets indicate the gained neutralization (%) by pre-incubation. As control, cells were incubated at each step with an equal volume of PBS or Y112R. Percentages of primary infection were calculated according to viral titers of PBS control conditions. *P* < 0.0001 (****) unless specified otherwise (2-tailed, unpaired Student's *t*-test). ns, not significant.

D   Inhibition of cell surface binding of SARS-CoV-2 VLP by FD20. VLP-GFP were preincubated, or not, with FD20 and RBM-targeting mAbs (CB6, REGN10933, REGN10987) for 30 min at 37°C. Then, the VLP premixes were incubated with VeroE6-hACE2 for 1 h at 37°C and the binding detected by flow cytometry. Control binding assays were performed using soluble ACE2 as a competitor. Data are representatives from three experiments.

Source data are available online for this figure.

and D) which represents the side views (Walls *et al*, 2020). This feature was also clearly visible in the 2-D class averages. By sharp contrast, this feature was lost upon FD20 incubation in both the raw images and the 2-D class averages (Fig 6E). As a control, S-2P incubated with an unrelated IgG (5E1) (Maun *et al*, 2010) did not show loss of integrity (Fig 6F). A time-course experiment showed that the S-2P protein started to lose integrity in the time window of 3–15 min upon FD20 incubation (Fig EV5). The results suggest that FD20 neutralizes SARS-CoV-2 mainly by destructing S. Because the FD20 epitope is structurally conserved in several other coronaviruses (Fig 4 F), their corresponding region may be exploited for the development of neutralizing antibodies with a similar mechanism.

# Discussion

Despite the successful development of vaccines, SARS-CoV-2 continues to threaten public health. Of great concern is the emergence of variants (such as the Delta variant) that are highly transmissible and capable of penetrating vaccine protection (Lopez Bernal *et al*, 2021). This highlights the need to develop broadly effective vaccines and mAbs. In this study, we discover a mAb (FD20) with broad activity against several Variants of Concern including the Delta owing to its conserved epitope.

FD20 joins CR3022 and EY6A as S-destructing (Huo *et al*, 2020; Zhou *et al*, 2020) neutralizing mAbs. This mechanism offers a

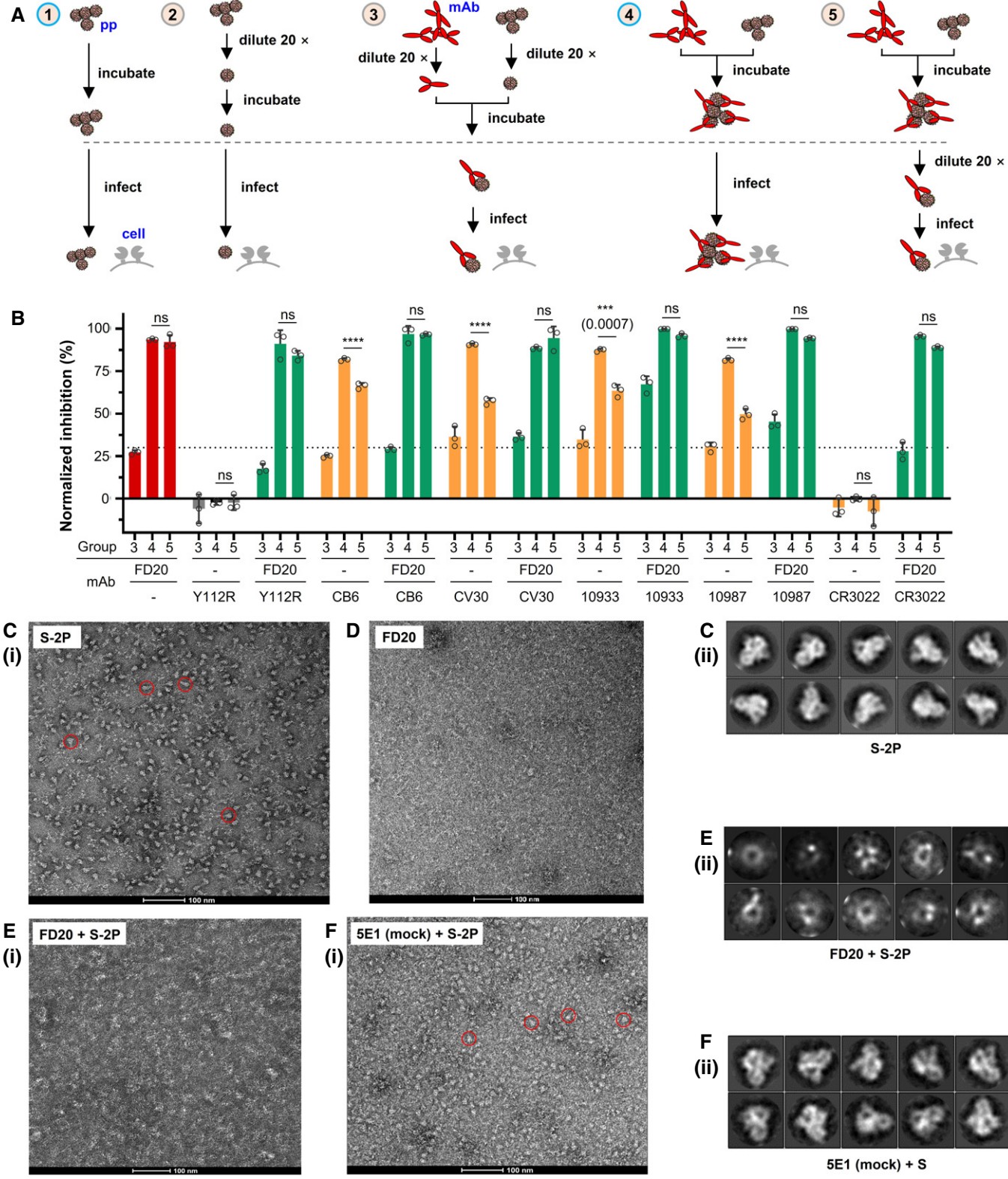

**Figure 6.**

**Figure 6. FD20 exhibits virucide effects before virus binding by destroying S trimer.**

A    Experimental design to evaluate the virucide effect of FD20, in the presence or not, of other antibodies. Different concentrations and 20-time dilution were used to achieve 70% inhibition when the antibody is provided alone. The comparisons of neutralization groups (3–5) relative to corresponding conditions without antibodies allow evaluating the irreversible effect of FD20 on particles. Group 4 is normalized using Group 1; groups 3 and 5 are normalized using Group 2.

B    FD20 acts irreversibly on particles in the presence or absence of other mAbs. As described in (A), 5 different conditions were used to evaluate the effect of SARS-CoV-2 pp pre-incubation with mAbs. In some experiments, FD20 was mixed with other mAbs and synergic effects were observed. Inhibition rate was calculated relatively to infection without antibodies at the similar SARS-CoV-2 pp input (diluted 20 times or not, Group 2/1 in A). Mean ± SD are plotted ($n = 3$ biological replicates). $P < 0.0001$ (****) unless specified otherwise (one-way ANOVA with Tukey's test); ns, not significant.

C–F    The destruction of SARS-CoV-2 S by FD20. (C) Negative-stain of FD20. (D-F) Negative-stain image (i) and 2D class averages (ii) of S alone (D), and S incubated with FD20 (E) or 5E1 (F) which is an unrelated mAb targeting a protein called Hedgehog (Maun et al, 2010). Bar = 100 nm. In D and F, red circles indicate typical side views of S. Results are representative of two independent experiments with two different S purification batches.

Source data are available online for this figure.

possible explanation for the apparent discrepancy between the cross-binding affinity and cross-neutralizing activity: FD20 binds to the RBDs from SARS-CoV and SARS-CoV-2 with comparable affinities but displays dramatically different neutralizing activities (Fig 1B–E). It has been reported that the S protein from SARS-CoV is more stable than that from SARS-CoV-2 (Ou et al, 2020). The higher stability of SARS-CoV S may be responsible for the higher resistance for FD20-mediated destruction and hence neutralization. This hypothesis remains to be tested using soluble S trimer from SARS-CoV.

Since the outbreak begins, thousands of neutralizing antibodies against SARS-CoV-2 have been reported (http://opig.stats.ox.ac.uk/webapps/coronavirus) and hundreds have their epitopes structurally characterized, most of which targets RBM and displays $IC_{50}$ values 1-2 orders of magnitude (Hansen et al, 2020; Shi et al, 2020) lower than FD20. For functional reasons, RBM is more exposed than other regions. Thus, the higher abundance of mAbs targeting the RBM is perhaps not unexpected. It is also possible that the non-RBM-targeting mAbs are generally less potent than RBM-targeting mAbs and are thus less prioritized for study and less reported. Notably, the $IC_{50}$ of FD20 (1.7 μg/ml) is similar to most of non-RBM-targeting mAbs (S2A4 (Piccoli et al, 2020), 3.5 μg/ml; H014 (Lv et al, 2020), 5.4 μg/ml; EY6A (Zhou et al, 2020), 0.07 and 20 μg/ml in two sets of experiments; CR3022 (Wu et al, 2020a), 5.2 μg/ml for SARS-CoV; CR3022-P384A (Wu et al, 2020a), 3.2 μg/ml for SARS-CoV-2), although more potent ones have also been reported (S309 (Pinto et al, 2020), 0.08 μg/ml; COVA1-16 (Liu et al, 2020), 0.02 μg/ml). Another possible reason for the relatively low neutralizing activity is its modest binding affinity ($K_D$ of 5.6 nM for the scFv form) for RBD. The structural information may be used to rationally design gain-of-function mutants to increase binding affinity and hence neutralizing activity, as demonstrated in our previous study with a neutralizing nanobody (Li et al, 2021). In vitro maturation methods such as soft randomization (Frei & Lai, 2016) and based-editor guided ex vivo maturation (Liu et al, 2018) may also be employed to increase affinity. Alternatively, because the unusual FD20 epitope does not overlap with most mAbs and nanobodies, fusing FD20 with high-affinity nanobodies may increase the apparent binding affinity by avidity effects (Yao et al, 2021).

The in vivo protection of FD20 was also modest. This could be due to the relatively low neutralization activity in vitro. Alternatively, it may be related to half-life which is not tested in this study. Given that hamsters are becoming a common animal model for SARS-CoV-2 infection and that little in vivo stability information is available for human mAb in hamsters, this issue warrants future investigation.

Mutagenesis data revealed, unlike RBM-targeting mAbs, little correlation between binding affinity and neutralization activity for FD20. This may be due to mechanistic differences. High affinity is required for RBM-targeting mAbs to persistently shield RBM from ACE2. On the contrary, the effect of FD20 should last even after its release if the S-destruction is irreversible. Therefore, neutralization by FD20 only requires affinity enough to outcompete the inter-protomeric interactions which are presumably not strong, as the S trimer is relatively unstable with a half-life of ~2 h at 42°C—a characteristic speculated to suit its high infectivity (Ou et al, 2020; Laporte et al, 2021). To some degree, FD20 mutants with fast off-rates may even be beneficial because of the possibility to dissociate from destructed S and become available for attacking the next S. This hypothesis remains to be carefully tested.

Several loss-of-function mutants on FD20's epitope are worth discussing. Although from natural SARS-CoV-2 isolates, W353R, R355T, D467Y all showed little or no infectivity (Table 2). These mutations either existed in minority variants that may or may not be infectious but were detected by sensitive sequencing techniques, or they require compensatory mutations for proper folding/function. For example, the isolate hCoV-19/USA/TX-HMH-MCoV-23597/2021 (www.gisaid.org) contains 34 other S mutations, including those in RBD such as A352 M and A372P besides R355T. Because simultaneous mutations happen less frequently than single mutations, it is less likely for them to evolve and escape FD20.

Finally, our work may inspire the design of broadly effective vaccines that stimulate antibodies against the FD20 epitope, for example, by engineering S proteins that preferably expose this region. FD20 could also be used as a tool to screen mAbs targeting this region by competitive binding assays.

## Conclusions

We have isolated a human neutralizing monoclonal antibody (FD20) against SARS-CoV-2 with modest neutralization activity in vitro and in vivo. By sharp contrast to several mAbs that are in (pre)-clinical trials, FD20 is resistant to SARS-CoV-2 mutations, showing no noticeable susceptibility to several known escape mutants and the variants B.1.1.7, B.1.351, P.1, and B.1.617.2 that have caused great public concerns recently. Biochemical and structural analysis of FD20-RBD reveals an uncommon neutralization mechanism by which the S is destroyed by FD20. Because FD20 recognizes an epitope that is distinctly different from most available RBM-targeting mAbs, it offers a basis for optimization to develop antibody mixes against SARS-CoV-2. Finally, the work validates a

previously predicted conserved and highly constrained epitope as a target for therapeutic antibodies that are less susceptible to escape mutants and hence may be of high clinical interest as it may neutralize quasispecies in patients.

# Materials and Methods

### S-RBD specific B-cell isolation from COVID-19 convalescent patients

The human-related study was approved by the Ethics Committee of Hangzhou Centre for Disease Control and Prevention (Approval No. 2020-7). All participants proved written informed consent before participation in this study and the experiments conformed to the principles set out in the WMA Declaration of Helsinki and the Department of Health and Human Services Belmont Report. All experiments with samples from COVID-19 convalescent patients were performed in a BioSafety Level 2$^+$ level laboratory and with approval from the Ethics Committee of Hangzhou Centre for Disease Control and Prevention (Approval No. 2020-7). Thirty COVID-19 convalescent patients were recruited and plasma RBD-binding titer was determined by ELISA. Blood samples from three convalescent patients with high plasma RBD-binding titer were drawn. Total B cells were isolated from COVID-19 convalescent patients' peripheral blood mononuclear cells using a human B cell isolation kit according to the manufacturer's instructions (Miltenyi). B cells from three convalescent patients were pooled (total 1.2 million B cells) and incubated with biotin-labeled SRAR-CoV-2 S-RBD protein, and S-RBD specific B cells were enriched using anti-biotin magnetic beads (Miltenyi). The isolated B cells, counted 5,000, were suspended in 1 ml TRIzol (Invitrogen), flash-frozen in liquid nitrogen, and stored in −80°C before RNA extraction.

### Construction of scFc library

Cells frozen in TRIzol were thawed at room temperature (22–25°C). Chloroform (200 μl) was added to the mixture, and the samples were vortexed before centrifugation for 10 min at 12,000× $g$ at 4°C. The aqueous phase was transferred to a new tube containing 500 μl chloroform and 1 μl of glycoblue (15 mg/ml). The resulting mixture was vortexed and centrifuged again for 5 min at 12,000× $g$ at 4°C. The aqueous phase was transferred to a new tube containing 2.5 volumes of ethanol to precipitate RNA. RNA pellets were then collected by centrifugation at 12,000× $g$ at 4°C for 20 min and dissolved in 20 μl of RNase-free H$_2$O. Reverse transcription PCR was performed with SuperScriptII RT (Invitrogen) with random hexanucleotides as primers, using a reaction program of 25°C for 10 min followed by 50°C for 50 min. The resulting cDNA was used for PCR amplification of Ig genes as follows.

V$_H$, V$_\lambda$, and V$_\kappa$ genes were amplified separately and combined by a two-step PCR. For each V gene family, a PCR was carried out using a forward V-gene primer and a reverse primer annealing to IgG, Igκ, or Igλ constant region. The PCRs were conducted in a 96-well plate with a total volume of 40 μl per well containing 2.5 μl cDNA, 1 μl 10 μM each primer, 20 μl of 2 × *TransTaq* High Fidelity (HiFi) PCR SuperMixII (TransGen Biotech). The reaction was initiated by heating at 94°C for 5 min, followed by 45 cycles of

denaturation (94°C 30 s), annealing (50°C 30 s for 2 cycles, 55°C for 15 cycles, and 60°C for 28 cycles), and extension (72°C 1 min), and ended with a final extension step at 72°C for 10 min. Nested PCR amplification consisted of a 25-μl reaction with 2 μl of the product from the first PCR, 1 μl of 10 μM each primer, and 12.5 μl of 2 × SuperMixII. The PCR conditions were the same as above except that the cycles were reduced to 40, and the annealing temperatures were set to 55°C for 15 cycles and 60°C for 25 cycles. The amplified products were purified from agarose gel for the V$_H$, V$_\lambda$, and V$_\kappa$ groups and used as templates for the assemble PCR with equal molar mixture of pooled V$_H$ with either V$_\lambda$ or V$_\kappa$ gene. The assembling PCR was performed for 7 cycles (94°C for 45 s, 60°C for 30 s, 72°C for 1 min) without primers, followed by an additional 25 cycles with the universal primers which anneal to the phage display vector. Final products were purified from agarose gel and ligated into the pDX-int vector by Type IIs restriction enzyme (BspQI)-mediated cloning. Primer sequences are listed in Appendix Table S2.

### Expression and purification—SARS-CoV-2 S receptor-binding domain (RBD)

The RBD was expressed in *Trichoplusia ni* High Five suspension cells (Thermo Fisher Cat. B85502, not tested for mycoplasma contamination) with the following polypeptide sequence, from N to C terminus: the honey bee melittin signal peptide KFLVNVALVFMVVYISYIYAA, a Gly-Ser linker, RBD (residues 330–531 of the UniProt P0DTC2), a Gly-Thr linker, the 3C protease recognition site (LEVLFQG^P), a Gly-Ser linker, the Avi tag for enzymatic biotinylation (GLNDIFEAQKIEWHE), a Ser-Gly linker, and a 10 × His tag for affinity purification. For purification, the medium containing secreted RBD was filtered through a 0.22-μm membrane, followed by batch-binding with Ni-Sepharose Excel resin (Cat 17-3712-03, GE Healthcare) in the presence of 20 mM imidazole. After 3 h, the beads were packed into a gravity column which was then washed with 10 column volume (CV) of 20 mM imidazole in **Buffer A** (150 mM NaCl, 20 mM Tris–HCl pH 8.0) before eluted with 300 mM imidazole in the same buffer. Biotinylation was carried out by incubating 0.8 mg/ml RBD with 22 μg/ml home-purified BirA (the biotinylation enzyme) in the presence of 5 mM ATP, 10 mM magnesium acetate, and 44 μM biotin for 16 h at 4°C. The mixture was subjected to gel filtration using a Superdex Increase 200 10/300 GL column. Fractions containing RBD were pooled, aliquoted, flash-frozen in liquid nitrogen, and stored at −80°C until use.

For crystallization, the RBD was treated with 3C protease (1:100 wt/wt, protease : RBD) to remove the C-terminal Avi- and His-tag. After protease digestion at 4°C for 16 h, the mixture was passed through a Ni-NTA column to remove His-tagged 3C. The flow-through fractions were pooled and concentrated to 8–10 mg/ml and mixed with 1.5-fold (molarity) of scFD20. The mixture was loaded to a Superdex Increase 200 10/300 GL column for gel permeation. Fractions containing the complex were pooled, concentrated to 7.6 mg/ml for crystallization.

RBD mutants for binding assays were generated by standard PCR-based site-directed mutagenesis and were produced essentially as the wild-type protein. Sequences of the mutants were verified by DNA sequencing.

### Expression and purification—scFv

Single-chain variable fragments (scFvs) were expressed in *E. coli* MC1061 cells with a pelB signal peptide (SKYLLPTAAAGLLLLAAQ-PAMA) at the N terminus, a GS linker (14 amino acids) between the two chains, and an Avi and a hexahistidine tag at the C terminus. Cells were first grown in Terrific Broth (TB, 1% (w/v) tryptone, 2% (w/v) yeast extract, 0.4% (v/v) glycerol in PBS buffer) at 37°C for 2 h to reach OD600 of ~0.5. The growth temperature was then lowered to 22°C for 1.5 h and the cells were induced with 0.02% (w/v) arabinose for 16 h at 22°C. Cells from 1 l of culture were then resuspended in 20 ml of hypertonic buffer (0.5 M sucrose, 0.5 mM EDTA, and 0.2 M Tris–HCl pH 8.0). After 0.5 h of dehydration, cells were rehydrated abruptly by adding 40 ml of MilliQ water. Cells were centrifuged and the supernatant which contains released scFv was incubated with Ni-NTA resin in the presence of 150 mM of NaCl, 2 mM of MgCl$_2$, and 20 mM of imidazole. After batch binding for 1 h at 4°C, the beads were packed into a gravity column, washed with 30 mM imidazole, and eluted with 300 mM imidazole in buffer containing 150 mM of NaCl and 20 mM Tris–HCl pH 8.0. For neutralization assays, the protein was desalted using a desalting column (Bio-Rad). For crystallization, the protein was directly mixed with RBD before being further purified by gel filtration.

### Expression and purification—IgG

DNA encoding the heavy and light chain variable regions was separately Gibson-assembled into a pDEC vector which contains the human IgG backbone. This construct allows the secretion of IgG into the culture medium. The two plasmids (1.4 mg/l for light chain, 0.6 mg/l for heavy chain) were co-transfected into 1 l of Expi293 cells (Thermo Fisher Cat. A14527, not tested for mycoplasma contamination) using polyethylenimine at a cell density of $2 \times 10^6$ per milliliter for transient expression. Valproic acid was added to a final concentration of 2 mM to aid expression. The medium containing secreted IgG-FD20 was collected 48–60 h post-transfection, filtered through a 0.22-μm membrane, and incubated with Protein A beads for 2 h at 4°C. The beads were packed into a gravity column and washed with 20 CV of PBS buffer, before eluted with acidic buffer containing 0.1 M glycine pH 3.0. The eluent was rapidly neutralized by adding with 1 M Tris–HCl pH 8.0. NaCl was added to a final concentration of 0.15 M. The purified FD20 was buffer-exchanged into PBS using a desalting column (Bio-Rad). FD20 was quantified using the theoretical molar extinction coefficient of 109,305/M/cm with absorbance measured using a Nanodrop machine.

### Expression and purification—SARS-CoV-2 S

The plasmid for mammalian transient expression of S was kindly provided by Prof. Yao Cong at the authors' institute. The pcDNA 3.1+ plasmid harbors the mammalian codon-optimized gene encoding residues Met1—Gln1208 of the SARS-CoV-2 S with mutations K986P, V987P, a GSAS linker substituting Arg682-Arg685 (the furin cleavage site), a C-terminal T4 fibritin trimerization motif (GYIPEAPRDGQAYVRKDGEWVLLSTFL), a TEV protease cleavage site, a FLAG tag, and a polyhistidine tag (Xu *et al*, 2020). For expression, 0.7 L of Expi293 cells (Cat. A14527, Thermo Fisher) at a density of $2 \times 10^6$ per milliliter were transfected with a mixture

containing 0.7 mg of plasmid and 1.4 mg of polyethylenimine. After 3.5 days of suspension culturing, the medium containing secreted S was collected, filtered through a 0.22-μm membrane, and adjusted to have 200 mM NaCl, 20 mM imidazole, 4 mM MgCl$_2$, and 20 mM Tris–HCl pH 7.5. The mixture was added with 3 ml of Ni-NTA beads and incubated at 4°C for 2 h for batch binding. The beads were packed into a gravity column, washed with 50 CV of 20 mM imidazole before eluted with 250 mM imidazole in 200 mM NaCl, 20 mM Tris–HCl pH 7.5. The eluted fractions containing S were pooled and buffer-exchanged into a buffer containing 200 mM NaCl, 20 mM HEPES pH 7.5 using a desalting column (Bio-Rad). Protein was concentrated using a 100-kDa cutoff membrane concentrator. SARS-CoV-2 S was quantified by measuring absorbance at 280 nm on a Nanodrop machine and calculated using a theoretical molar extinction coefficient of 138,825/M/cm.

### Phage display

Two rounds of phage display were performed as follows. For the first round, phage particles were incubated with 50 nM of biotinylated RBD, before added into a 96-well plate that had been coated with 60 nM neutravidin (Cat. 31000, Thermo Fisher Scientific). The plate was washed with **Buffer B** (150 mM NaCl, 0.05% (w/v) Tween 20, 20 mM Tris–HCl pH 7.4), and the bound phage particles were released from the plate by tryptic digestion with 0.25 mg/ml trypsin in the buffer containing 150 mM NaCl and 20 mM Tris–HCl pH 7.4. The enriched phage particles were amplified, incubated with 50 nM biotinylated RBD, before selected by another round using 12 μl of MyOne Streptavidin C1 beads. To compete off phage particles expressing fast off-rate binders, the mixture was incubated with 5 μM of non-biotinylated RBD for 3 min. The beads were washed with Buffer B, and the remaining binders were released by tryptic digestions. The resulted phagemids were extracted and sub-cloned into pSb_init vector (Zimmermann *et al*, 2018) by fragment-exchange (FX) cloning and transformed into *E. coli* MC1061 for further screening at a single-colony level (Zimmermann *et al*, 2018, 2020).

### Identifying RBD-binders by ELISA

Single colonies carrying plasmids (pSb-init) (Zimmermann *et al*, 2018) harboring different scFvs were inoculated into 96-well plates. Cells were grown for 5 h at 37°C before diluted (1:20) into 1 ml of TB with 25 μg/ml chloramphenicol. Cells were induced with 0.02% (w/v) arabinose at 22°C for 17 h before collected by centrifugation at 3,220 g for 30 min. The biomass was resuspended in TES Buffer (20% (w/v) sucrose, 0.5 mM EDTA, 0.5 μg/ml lysozyme, 50 mM Tris–HCl pH 8.0) and incubated with gentle shaking for 30 min at room temperature (RT, 22–25°C). The lysate was added with 1 ml of **TBS** (150 mM NaCl, 20 mM Tris–HCl pH 7.4) supplemented with 1 mM MgCl$_2$. The mixtures were clarified by centrifugation at 3,220 *g* for 30 min at 4°C. The supernatant which contains scFv was used directly for ELISA or FSEC screening (below).

For ELISA, a Maxi-Sorp plate 96 well (Cat. 442404, Thermo Fisher) was coated with Protein A at 4°C for 16 h. Unbound Protein A was removed and the plate was blocked with 0.5% (w/v) bovine serum albumin (BSA) solubilized in **TBS** buffer at RT. After 30 min, the plate was washed three times by **TBS** followed by incubation with anti-Myc antibodies (Cat. M4439, Sigma, 1:2,000 dilution) in a

TBS-BSA-T buffer (TBS with 0.05% (v/v) Tween 20 and 0.5% (w/v) BSA). After 20 min of incubation at RT, the plate was washed three times with **TBST** (TBS supplemented with 0.05% Tween 20) to remove excess antibodies. Myc-tagged scFv (in the periplasm extract, see above) was added to the plate and incubated for 20 min. After washing three times using **TBST**, biotinylated RBD or MBP (maltose-binding protein) was added to each well to 50 nM. After 20 min of incubation, the solution was removed and the plate was washed three times using **TBST**. The wells were added with streptavidin-conjugated with horseradish peroxidase (HRP) (1:5,000, Cat S2438, Sigma). After 20 min of incubation, the plate was rinsed three times using **TBST**. To develop ELISA signal, 100 µl of developing buffer (51 mM $Na_2HPO_4$, 24 mM citric acid, 0.006% (v/v) $H_2O_2$, 0.1 mg/ml 3,3',5,5'-tetramethylbenzidine) were added to each well. $A_{650}$ was recorded using a plate reader.

### Fluorescence-detector size exclusion chromatography (FSEC)

Biotinylated RBD was mixed with streptavidin (Cat 16955, AAT Bioquest) that was labeled with fluorescein. Either purified scFv or periplasmic extract was mixed with the fluorescently labeled RBD for fluorescence-detector size exclusion chromatography (FSEC). The final concentration of RBD was at 500 nM. FSEC was run with a Sepax analytic gel filtration column connected to a Shimadzu HPLC equipped with a fluorescence detector. Fluorescence with excitation and emission wavelength of 482/508 nm was monitored.

### Bio-layer interferometry assay

The binding between scFv (scFD20) or IgG (FD20) and RBD (SARS-CoV-2 RBD and mutants, purified and biotinylated in this study; biotinylated SARS-CoV RBD, Cat. 40150-V08B2-B, Sino Biological) were monitored using a bio-layer interferometry (BLI) assay with an Octet RED96 system (ForteBio). Biotinylated RBD (2 µg/ml) was immobilized onto a streptavidin SA sensor (Cat 18-5019) in the presence of **Buffer C** (0.05% (v/v) Tween 20 in PBS buffer) at 30°C. The binding was allowed to equilibrate (baseline) for 60–120 s, before binding with scFD20 or FD20 for 360 s at various concentrations as indicated in individual legends (association). The dissociation phase was monitored by soaking the sensor in **Buffer C** for 600 s. Binding kinetics parameters for scFv-RBD interactions were obtained by global fitting using a 1:1 stoichiometry with the built-in software Data Analysis 10.0. For IgG-RBD interactions, bridged complex may form between the bivalent analyte (IgG) and closely spaced RBD. The resulting apparent affinities are less informative and hence are not reported.

To assess if the binding between RBD and ACE2 is inhibited by scFD20, the sensor immobilized with RBD was soaked in **Buffer C** containing 200 nM of scFD20, and BLI signal was monitored in the presence or absence of ACE2 at 25 nM. As a control, the ACE2-RBD binding progress curve was obtained using the sensor treated with scFD20-free buffer.

### Pseudotyped retroviral particle production, infection, and neutralization assays

All experiments with SARS-CoV-2 pseudovirus particles were performed in a P2 level laboratory and with approval from the Institut Pasteur of Shanghai, Chinese Academy of Sciences. The retroviral pseudotyped particles were generated by co-transfection of HEK293T cells (Cat. CRL-3216, ATCC, tested for free of mycoplasma contamination) using polyethylenimine with the expression vectors encoding the various mutants of SARS-CoV-2 S (Wuhan-Hu-1, GenBank: QHD43419.1), or S from Variants of Concern (Lineage B.1.1.7, isolate hCoV-19/England/204820464/2020; Lineage B.1.351, isolate hCoV-19/USA/MD-HP01542/2021; Lineage P.1, isolate hCoV-19/Japan/TY7-503/2021; Lineage B.1.617.2, isolate hCoV-19/USA/PHC658/2021); or SARS-CoV (isolate Frankfurt-1 FFM-1) truncated viral envelope glycoproteins, the murine leukemia virus core/packaging components (MLV Gag-Pol), and a retroviral transfer vector harboring the gene encoding the green fluorescent protein (GFP). Note that the SARS-CoV-2 S variants were all truncated by removing the 19 amino acids at the C terminus. Supernatants that contained pseudotyped particles were harvested 48 h post-transfection and filtered through a 0.45-µm membrane before being used for infection assays.

To evaluate the mAbs neutralization activity against MLV pseudotyped viruses, VeroE6-hACE2 cells (Li *et al*, 2021) (VeroE6 cells: Cat. CRL-1586, ATCC, tested for free of mycoplasma contamination) were seeded at 10,000 cells/well in 48-well plates using DMEM supplemented with 10% FBS, 1% penicillin–streptomycin. After 24 h, the serial dilutions of mAbs alone, or in combination (1:1 ratio) in 50 µl DMEM were mixed with 100 µl pseudotyped viruses in different conditions described in figure legends. After infection 6 h at 37°C, cell media were changed and further incubated for 48 h before cells were analyzed by fluorescence-activated flow cytometer. By comparing to the infectivity of MLV pseudotyped viruses incubated with DMEM containing 2% fetal calf serum (FBS) (which was standardized to 100%), the neutralization activity of serially diluted mAbs will be calculated.

Mutant SARS-CoV-2 pp were generated by replacing the wild-type S with mutations generated by standard PCR-based mutagenesis. Sequences were verified by DNA sequencing.

### Cell–cell fusion assay

HEK293T cells ($2.5 \times 10^5$ cells/well seeded in 6-well plate 24 h before transfection) were co-transfected with plasmids of SARS-CoV-2 Spike protein (0.4 µg) and Loxp-Cre recombinase (2 µg) per well. In parallel, the plasmids encoding hACE2 and Loxp-stop-Luc were co-transfected into HEK293T cells in another 6-well plate at the same time. After 24-h transfection, these transfected cells were detached with versene (0.02% EDTA; Thermo Fisher) and co-cultured at 1:1 ratio in 96-well plates at 10,000 cells/well in the presence, or not, of 10-fold serial diluted mAbs. After 24-h incubation at 37°C, cells were lysed by adding 20 µl lysis buffer with the substrates. The luminescence signal was recorded with Synergy H1 Hybrid Multi-Mode plate reader. The percentage of fusion inhibition is calculated by comparing to fusion without mAb (which was standardized to 100%).

### Kinetic of action assay

VeroE6-hACE2 cells (Li *et al*, 2021) were incubated with MLV pseudotyped viruses (SARS-CoV-2 pp) during binding (1 h at 4°C), entry (6 h at 37°C), or infection (1 h at 4°C and 6 h at 37°C), mAbs FD20,

Y112R, CB6, CV30, REGN10933, REGN10987, CR3022 (25 nM) were also added, or not, onto cells during the different infection conditions. After 6-h infection, remove the inoculum and change to the fresh media for later 48-h incubation at 37°C. As a control, cells were incubated at each step with an equal volume of DMEM along with SARS-CoV-2 pp, and the percentages of its infection were quantified at 48 h post-infection and standardized to 100%.

### Infection-dilution assay

Based on the $IC_{50}$ curve calculated by neutralization assays of each mAb, the concentrations of each mAb were adjusted to the neutralization ability around 30% against SARS-CoV-2 pseudotyped viruses, which were 2.5 nM for FD20, 1.25 nM for CV30, 0.25 nM for CB6, REGN10933 and REGN10987, 2.5 nM for Y112R and CR3022 as a control. SARS-CoV-2 pp was pre-incubated with 50 nM FD20 (as well as 25 nM CV30, 5 nM CB6, 5 nM 10933, 5 nM 10987, 50 nM Y112R, 50 nM CR3022, separately) at 37°C for 1 h, and then diluted for 20 fold, or not, with fresh medium to reach suboptimal mAb concentration (30%) As a control, diluted SARS-CoV-2 pp was pre-incubated with 2.5 nM FD20,1.25 nM CV30, 0.25 nM CB6, 0.25 nM REGN10933, 0.25 nM REGN10987, 2.5 nM Y112R or 2.5 nM CR3022 at 37°C for 1 h prior to inoculation.

For mAb cocktail dilution, the concentration described above were used for each mAbs; leading to 1:1 equimolar ratio for FD20 + Y112R, FD20 + CR3022, 2:1 molar ratio for FD20 + CV30, 10:1 molar ratio for FD20 + CB6, FD20 + REGN10933, FD20 + REGN10987. Cocktails were pre-incubated with SARS-CoV-2 pp, then followed, or not, by 20-fold dilution in the fresh medium. The 20-fold diluted mAb cocktails were also pre-incubated SARS-CoV-2 pp as the control.

All infections were conducted on 10,000 VeroE6-hACE2 cells seeded in a 48-well plate, and the infection was analyzed by flow cytometry at 48 h post-infection.

### Binding assay using GFP-VLPs

For the production of GFP-tagged VLP, the plasmids pVAX-SARS-CoV-2 S, pCDNA3.1(+)SARS-CoV-2 M, pCDNA3.1(+)SARS-CoV-2 N, pCDNA3.1(+)SARS-CoV-2 E were transfected in HEK293T cells in S:M:N:E = 1:1:2:2 molar ratio using PEI transfection protocol. After 48 h and 72 h, the cell culture supernatant was collected and GFP-VLPs concentrated by centrifugation at 40,000 rpm for 3.5 h through a 20% sucrose cushion. The pellet containing GFP-tagged VLP was dissolved in 100 µl PBS and stored in −80°C.

For binging assay, VeroE6-hACE2 cells were collected using EDTA and washed twice in ice-cold PBFA (PBS-2% FCS-0.1% Azide). Concentrated GFP-VLP, preincubated or not with 30 nM of different antibodies for 30 min at 37°C, were bound onto $2 \times 10^5$ cells for 1 h at 37°C. Cells were washed twice in PBFA and binding was then measured by flow cytometry.

### Western blot for the expression and processing of S and S variants in SARS-CoV-2 pp

HEK293T cells were seeded as $2.5 \times 10^5$ cells/well in 6-well plate 24 h before transfection. To generate the retroviral pseudotyped particles with S and S mutants, the plasmid encoding the S variant

(without the 19 amino-acids at the C terminus), the murine leukemia virus core/packaging components (MLV Gag-Pol), and a retroviral transfer vector harboring the gene encoding the green fluorescent protein (GFP) were combined for co-transfection of HEK293T cells using polyethylenimine (PEI). Supernatants containing the pseudotyped particles were harvested 48 h post-transfection and filtered through a 0.45-µm membrane. The cell lysates were collected in 150 µl NP-40 lysis buffer (Beyotime) on ice for 15 min for each well after being washed by ice-cold PBS buffer. Lysates were collected and centrifuged at 4°C for 10 min at $16,000 \times g$. Both the supernatants of cell lysates and the collected supernatants containing pseudotyped particles were boiled in denatured gel sample loading buffer (Solarbio) and analyzed by SDS–PAGE. Proteins were transferred onto polyvinylidene difluoride (PVDF) membranes, and the blots were blocked with 5% milk in Tris-buffered saline with 0.1% Tween (TBST). Immunoblots were incubated with primary antibodies against SARS-COV-2 S trimer (Mouse, Cat. 40591-MM42, Sino Biological, Beijing, China) (1:1,000 dilution with 5% milk in TBST). For control purposes, the same blots were incubated with antibodies against GAPDH (glyceraldehyde-3-phosphate dehydrogenase) (Mouse, Cat. 60004-1-Ig, Proteintech, Wuhan, China) (1:2,000 dilution with 5% milk in TBST). The incubation for primary antibodies was performed for overnight at 4°C. The blots were washed three times with TBST for 40 min each time and then incubated for 30 min with horseradish peroxidase-linked anti-mouse secondary antibodies (Goat, Cat. SA00001-1, Proteintech) diluted 1:2,000 in milk-TBST. After washing, the positive bands were detected after the addition of chemiluminescence reagents and visualized on a Tanon 4600SF machine.

### Neutralization assay using authentic SARS-CoV-2 virus (hCoV-19/China/CAS-B001/2020)

Neutralization assays using the SARS-CoV-2 strain hCoV-19/China/CAS-B001/2020 were performed in a biosafety level 3 laboratory with approval from the Institute of Microbiology, Chinese Academy of Sciences. A microneutralization assay was used to determine the neutralizing activity of FD20 against live SARS-CoV-2 virus (hCoV-19/China/CAS-B001/2020, National Microbiology Data Center *NMDCN0000102-3*, GISAID databases *EPI_ISL_514256-7*; or BetaCoV/Munich/BavPat1/2020; or the UK/B.1.1.7 variant). Purified FD20 with serial dilutions were mixed with an equal volume of viral suspension containing 2,000 median tissue culture infective dose ($TCID_{50}$) virus titers per milliliter. The antibody–virus mixture was incubated for 1 h at 37°C. After incubation, 100 µl of the mixture at each dilution was added in octuplicates into a 96-well plate containing a semi-confluent VeroE6 monolayer. The plates were incubated for 3 days at 37°C followed by inspection under an inverted optical microscope for cytopathic effect (CPE). $IC_{50}$ was determined by curve fitting using GraphPad Prism 8.

### Plaque-reduction neutralization using authentic SARS-CoV-2 (strains BetaCoV/Munich/BavPat1/2020 and UK/B.1.1.7)

Neutralization assays using SARS-CoV-2 strains BetaCoV/Munich/BavPat1/2020 and UK/B.1.1.7 were performed in a biosafety level 3 laboratory with approval from the Erasmus Medical Center. An in-house gold standard plaque-reduction neutralization assay (PRNT)

was used as a reference as previously described by Okba *et al* (2020). The virus strains BetaCoV/Munich/BavPat1/2020 (400 plaque-forming units) were pre-incubated with serially diluted mAbs at 37°C for 1 h before placing the mixtures on Vero-E6 cells. After incubation for 1 h and wash, cells were fixed after 2 days with 4% formaldehyde in PBS and stained with polyclonal rabbit anti-SARS-CoV-2 antibodies (Cat. 40589-T62, Sino Biological, Beijing, China; 1:1,000 dilution). After a secondary peroxidase-labeled goat anti-rabbit IgG (Cat. P0448, Agilent Dako; 1:100 dilution) incubation, the foci were colored using a precipitate-forming 3,3′,5,5′-tetramethylbenzidine substrate (True Blue; Kirkegaard and Perry Laboratories) and counted to measure neutralization rate.

## Crystallization, data collection, processing, and structure determination

For crystallization, 100 nl of scFv-RBD complex was mixed with an equal volume of precipitant solution using a Gryphon LCP robot in a two-well sitting drop plate with 70 µl of reservoir solution. Crystals of FD20 grew in the precipitant solution containing 0.2 M lithium sulfate, 25% (w/v) polyethylene glycol 3,350, 0.1 M Bis-Tris pH 5.5. Crystals aged 10 days were harvested, transferred stepwise to 15% (v/v) glycerol in the mother liquor for cryo-protection before flash-cooled in liquid nitrogen. Diffraction data were collected on a Pilatus 6 M detector at beamline BL18U1 at the Shanghai Synchrotron Radiation Facility using a $50 \times 50$ µm X-ray beam with a wavelength of 0.97853 Å. Data were integrated with XDS (Kabsch, 2010) and scaled and merged using Aimless (Evans & Murshudov, 2013). The structure was solved by molecular replacement using Phaser (McCoy *et al*, 2007) with the appropriate part of PDB entry 6M0J (Lan *et al*, 2020) and 5C6W as search models. The model was adjusted manually in Coot (Emsley *et al*, 2010) using $2F_o\text{-}F_c$ maps and refined using Phenix. Structures were visualized using PyMOL (PyMOL, 2015).

## Negative-stain electron microscopy and 2D classification

SARS-CoV-2 S trimer (S-2P, see S purification) at 0.2 mg/ml was either incubated with PBS buffer or with FD20 at a molar ratio of 1:6 (S trimer : IgG) at 4°C for 1 h. The samples were then diluted to 40 µg/ml and applied to a copper grid that has a supporting carbon film (Cat BZ31024a, Beijing Zhongjingkeyi Technology) which had been treated by $H_2$ and $O_2$ for 20 s in a plasma cleaner (Model 950 Advanced Plasma System, Gatan). The grid was then stained with 5% (w/v) uranyl acetate. The samples were imaged using a Tecnai G2 Spirit electron microscopy (Thermo Fisher) operated at 120 kV. Electron microscopic data were collected on a $4K \times 4K$ CCD Eagle camera, at a nominal magnification of 67,000, which corresponds to a physical pixel size of 1.74 Å.

Negative staining particles were automatically picked using EMAN 2.2 (Tang *et al*, 2007). Particles were imported to Relion 3.1 (Scheres, 2012) for 2D classification. A total number of 7,740 particles from 22 micrographs were used for the 2D classification (K = 30) of the S alone; 4,714 particles from 21 micrographs were used for the 2D classification (K = 20) of FD20-treated S.

For time-course study, the same procedure was followed except that negative-staining samples were prepared for S-2P (i) without FD20 incubation, (ii) immediately after mixing (< 0.5 min), (iii)

with 3 min of FD20 incubation, and (iv) with 15 min of FD20 incubation.

## Animals and ethical statement

All animal experiments with live SARS-CoV-2 viruses were performed in an ABSL3 biocontainment laboratory. The research was conducted in compliance with the Dutch legislation for the protection of animals used for scientific purposes (2014, implementing EU Directive 2010/63) and other relevant regulations. The licensed establishment where this research was conducted (Erasmus MC) has an approved OLAW Assurance # A5051-01. The research was conducted under a project license from the Dutch Central Commission on Animal experiments (CCD), and the study protocol (#17-4312) was approved by the institutional Animal Welfare Body. Animals were socially housed (2–3 animals per filter top cage, (T3, Techniplast), placed in Class III isolators, under controlled conditions of humidity (55%, range: 50–60%), temperature (21°C, range: 19–23°C), airflow in isolator (30 m³/h, range: 25–35 m³/h) and light regime (12-h light/12-h dark cycles). Food and water were available ad libitum. Animals were cared for and monitored (pre- and post-infection) by qualified personnel. The animals were sedated/anesthetized for all invasive procedures.

## Animal procedures

Female Syrian golden hamsters (*Mesocricetus auratus*; strain RjHan: AURA, purpose bred from Janvier, France) were allowed to acclimatize and aged 6 weeks. Animals were randomly divided into experimental and control groups. For procedures, they were briefly anesthetized by chamber induction (5 l 100% $O_2$/min and 3–5% isoflurane); 6 h before inoculation with the virus, groups of 5 animals were treated with 5 mg of FD20 in 0.5 ml via the intraperitoneal route.

Animals were inoculated with $10^5$ $TCID_{50}$ of SARS-CoV-2 (isolate BetaCoV/Munich/BavPat1/2020) or PBS (mock controls) in a 100 µl volume via the intranasal route. Animals were monitored for general health status and behavior daily and were weighed regularly for the duration of the study (up to 4 days post-inoculation; d.p.i.). On 4 d.p.i. groups of 5 animals were euthanized and lung, and nasal turbinates, were removed for virus detection and histopathology.

## Virus detection

Samples from nasal turbinates and lungs were collected post mortem for virus detection by RT–qPCR and virus isolation as previously described (Rockx *et al*, 2020). The SARS-CoV-2 RT–qPCR was performed and quantified as $TCID_{50}$ equivalents using a standard curve of the virus stock as previously published (Corman *et al*, 2020). Mean and standard error of the mean of all five individual data points were reported. For virus titer, the raw data were log-transformed to approximate normal distribution before statistical analysis using 2-tailed, unpaired Student's *t*-test.

## Histopathology and immunohistochemistry

For histological examination, lung and nasal turbinates were collected. Tissues for light microscope examination were fixed in

**The paper explained**

**Problem**

SARS-CoV-2 has caused a global pandemic with more than 200 million confirmed infections so far and the efficacy of the current vaccines is greatly challenged by the emergence of Variants of Concern (VOCs) such as the Delta. Monoclonal antibodies (mAbs) remain to be an important therapeutic asset for COVID-19 but they also face the challenge of being escaped by VOCs. The situation calls for developing broadly active mAbs that neutralize VOCs but reports on such antibodies are relatively rare.

**Results**

We have obtained a broadly active mAb (FD20) that neutralizes SARS-CoV-2 and several known escape mutants and four current VOCs including the most destructive Delta. In addition, it shows mild cross-activity against SARS-CoV. Prophylactic injection provides protection of hamsters from SARS-CoV-2 infection. Structural studies show a neutralization mechanism through which FD20 disassemble the viral surface protein Spike by engaging at a conserved region. Mutagenesis studies show that FD20 remains active against naturally occurring and laboratory-made mutations on the FD20 epitope, suggesting a low likelihood for escape mutants to evolve.

**Impact**

Our results reveal a highly conserved vulnerability site in SARS-CoV-2 for developing therapeutic antibodies and may inspire rational design to engineer vaccines with the conserved site more exposed to elicit broadly active antibodies. Owing to its broad neutralization activity and its potential to inhibit syncytia, FD20 is a promising candidate for further optimization as a therapeutic mAb. Because FD20 recognizes an unusual epitope, it can be used as a component in antibody cocktail together with existing ultra-potent, non-competing mAbs.

10% neutral-buffered formalin, embedded in paraffin, and 3-μm sections were stained with hematoxylin and eosin. Sections of all tissue samples were examined for SARS-CoV-2 antigen expression by immunohistochemistry as previously described (Rockx *et al*, 2020). Briefly, rehydrated sections were heated in a citrate buffer (pH 6.0) at 100°C for 15 min, before being treated with 3% $H_2O_2$ to block background peroxidase activity. Slides were washed with PBS containing Tween 20 and blocked with goat serum (10%) at room temperature for 30 min, before being incubated with the anti-necleoprotein (SARS-CoV-2) polyclonal antibody (rabbit, Cat. 40143-T62, Sino Biological, Chesterbrook, PA, USA; 1:1,000 dilution) in PBS buffer plus 0.1% BSA. After washing, the slides were incubated with goat anti-rabbit IgG conjugated with the horseradish peroxidase (Cat. P0448, DAKO, Agilent Technologies Netherlands, The Netherlands; 1:100 dilution) for 1 h at room temperature. Horseradish activity was developed using 3-amino-9-ethylcarbazole for 10 min, and the sections were counterstained with hematoxylin. For quantitative assessment of SARS-CoV-2 infection-associated inflammation in the lung, each H&E-stained section was examined for inflammation by light microscopy using a 2.5× objective, and the area of visibly inflamed tissue as a percentage of the total area of the lung section was estimated. Quantitative assessment of virus antigen expression in the lung was performed according to the same method, but using lung sections stained by immunohistochemistry for SARS-CoV-2 antigen. Sections were examined without knowledge of the identity of the hamsters.

## Statistical analyses

Data were plotted as mean ± SD or SEM as specified in the caption with $n$ indicating either biological replicates for cell-based neutralization assays or number of animals for *in vivo* virus challenge experiments. Biological replicates of mostly 3 or 4 (see exact $n$ in figure legends), and animal size of 5 were chosen based on our previous experience. Exact $P$ values were included in the figure except for cases where ****$P < 0.0001$. Statistical analysis was performed using 2-tailed, unpaired Student's $t$-test for comparisons between two group, and using one-way ANOVA followed by Tukey's test for comparison among three or more groups. Data from biological replicates in cell-based neutralization assays were assumed to follow normal distribution without normality test owing to the small numbers of replicates. Data for virus titer were log-transformed to approximate normal distribution before statistical analysis. Hamsters were randomly divided into experimental and control groups. No data were excluded during analysis. Pathological analysis was performed without knowing the identity of the experimental/control groups. No blinding was applied to other analyses (virus titer and body weight).

## Data availability

The datasets produced in this study are available in the following databases: structure factors and coordinates: Protein Data Bank 7CYV (https://www.rcsb.org/structure/7CYV). The raw data for Table 2, Figs 1D , 2, 3, 5C and 6B are provided as source data. The uncropped images of Fig EV2E and F are provided as source data.

Expanded View for this article is available online.

## Acknowledgements

We thank Prof. Yao Cong at the authors' institute for providing the S plasmid, and Dr. Yanxing Wang and Mr. Yitian Luo for technical assistance on negative staining. We thank the staff members of the Large-scale Protein Preparation System for equipment maintenance and management, at the SSRF-BL18U1 and BL19U1 beamlines at National Facility for Protein Science (Shanghai) and of the electron microscope facility for technical support and assistance. This work has been supported by the Strategic Priority Research Program of CAS (XDB37020204, D. Li; XDB29010102, Y.B.), Key Program of CAS Frontier Science (QYZDB-SSW-SMC037, D. Li), CAS Facility-based Open Research Program (2017), the National Natural Science Foundation of China (NSFC) (31870726, D. Li; 31870153, D. La; 31970880, F.L.M), Ministry of Science and Technology of China (2020YFC0845900, D. La.; 2017YFA0506700, F.L.M), CAS President's International Fellowship Initiative (2020VBA0023, DLa), the Key R & D Program of Jiangsu Province (Social Development) Project (BE2019625, DLa), Science and Technology Commission of Shanghai Municipality (STCSM) (20ZR1466700, D. Li; 20490760200, F.L.M), Shanghai Municipal Science and Technology Major Project (20431900402, D. La.), Innovation Capacity Building Project of Jiangsu province Nanjing Unicorn Academy of innovation (BM2020019, D. La.), the ERINHA-Advance project (funding from the European Union's Horizon 2020 Research & Innovation program, grant agreement No. 824061), and Science and Technology Commission of Hangzhou Municipality (202013A02, J.C.). This project is included in RECOVER European Union's Horizon 2020 research and innovation program under grant agreement No 101003589. Y.B. is supported by the NSFC Outstanding Young Scholars (31822055) and Youth Innovation

Promotion Association of the Chinese Academy of Sciences (Youth Innovation Promotion Association CAS) (2017122). F.L.M is supported by a CAS grant (JCTD-2020-17).

## Author contributions

TL, YLi, and YLa isolated, characterized, and crystalized scFv-RBD complexes with guidance from HC and HY. HC, YLa and TL expressed and purified IgG and SHC performed negative staining. HY and TL performed binding assays. YZ, BZ, LL, SX, MM, and BD did neutralization assays under the supervision of DLa. LDL and YW constructed phage display library with assistance from TL and under the supervision of FLM. NZ and YG worked on live virus under supervision of YB. TK performed histopathological analysis. CGvK performed virus neutralization. MFvV and ASR designed animal studies. BR designed and coordinated animal studies. ZS, XZ, JL performed patient profiling under the supervision of JC. JF, LW CX, HZ, and FLM developed reagents and performed enrichment of B cells. DLi and WQ collected diffraction data. DLi determined and analyzed the structure. DLi and FLM initiated the project. Z.H. provided reagents for neutralizing assays. DLi and DLa wrote the manuscript with inputs from TL, HY, HC, MFvV, BR, ASR, HR, and FLM.

## Conflict of interest

A patent application for monoclonal antibody therapy for the treatment of COVID-19 has been filed for FD20 with application No. 202011407906.7 (China National Intellectual Property Administration).

## For more information

A   COVID-19 Dashboard by the World Health Organization (WHO): https://covid 19.who.int/

B   Variants of Concern tracker by WHO: https://www.who.int/en/activities/tracking-SARS-CoV-2-variants/

C   Sequence and epitope information about antibodies against coronaviruses: http://opig.stats.ox.ac.uk/webapps/covabdab/

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
