## [Review Process File · EMBO Molecular Medicine]

Uncovering a conserved vulnerability site in SARS-CoV-2 by a human antibody

Tingting Li, Hongmin Cai, Yapei Zhao, Yanfang Li, Yanling Lai, Hebang Yao, Liu Liu, Zhou Sun, Martje Vlissingen, Thijs Kuiken, Corine GeurtsvanKessel, Ning Zhang, Bingjie Zhou, Lu Lu, Yuhuan Gong, Wenmin Qin, Moumita Mondal, Bowen Duan, Shiqi Xu, Audrey Richard, Hervé Raoul, JianFeng Chen, Chenqi Xu, Ligang Wu, Haisheng Zhou, Zhong Huang, Xuechao Zhang, Jun Li, Yanyan Wang, Yuhai Bi, Barry Rockx, Junfang Chen, Feilong Meng, Dimitri LAVILLETTE, and Dianfan Li

DOI: [10.15252/emmm.202114544](https://doi.org/10.15252/emmm.202114544)

Corresponding authors: Dianfan Li (dianfan.li@sibcb.ac.cn), Dimitri LAVILLETTE (dlaville@ips.ac.cn), Feilong Meng (Feilong.Meng@sibcb.ac.cn), Junfang Chen (hzjkcjf@163.com)

Review Timeline:

Submission Date:	12th May 21
Editorial Decision:	4th Jun 21
Revision Received:	15th Sep 21
Editorial Decision:	6th Oct 21
Revision Received:	14th Oct 21
Accepted:	18th Oct 21

Editor: Zeljko Durdevic

Transaction Report:

4th Jun 2021

Dear Dr. Li,

Thank you for the submission of your manuscript to EMBO Molecular Medicine. We have now received feedback from the three reviewers who agreed to evaluate your manuscript. As you will see from the reports below, the referees acknowledge the interest of the study but also raise important critique that should be addressed in a major revision.

We would welcome the submission of a revised version within three months for further consideration. However, we realize that the current situation is exceptional on the account of the COVID-19/SARS-CoV-2 pandemic. Please let us know if you require longer to complete the revision.

I look forward to receiving your revised manuscript.

Yours sincerely,

Zeljko Durdevic

**** Reviewer's comments ****

Referee #1 (Remarks for Author):

The manuscript by T. Li et. al. reports isolation and characterization of a non-RBM monoclonal antibody (FD20) from COVID-19 convalescent patients with good neutralization activity against SARS-CoV-2. Although many neutralized Abs for SAR-CoV-2 have been reported before, the importance of FD20 is that it targets to a conserved epitope and neutralizes SARS-CoV-2 including the current concerning lineages B.1.1.7 and B.1.351. The authors have done experiments from different angles to prove the activities and potential mechanisms. I only have some minor concerns:

1. Fig. 1A : For the screening, the authors mentioned that B cells from three convalescent patients were pooled. Are there any criteria for choosing the convalescent patients? At the end of the screening, how many neutralized antibodies were obtained besides scFD20?
2. Line 184 We next investigated the in vivo potential of FD20 by intraperitoneal injection 6 h before intranasal inoculation of hamsters by live viruses (Figure 3): Is this sentence meaning that FD20 was injected first and then virus infection after 6h? It looks strange since normally for antibody treatment, the animals should be infected with virus first and then inject antibody for treatment.
3. Fig. 3B and 3E : The virus titers in either the lung or the nasal tissues determined by PCR and TCID50 were not consistent. For example, there are two lung tissues showing 0 TCID50 (Fig. 3C) and the PCR results (Fig. 3B) are quite high. What could be the reasons behind? It is quite hard to understand why the titers were varied so much between the PCR method and the TCID50 method.

Referee #2 (Remarks for Author):

COVID-19, the pandemic caused by SARS-CoV-2, is still threatening thousands of millions of people around the world. In the absence of specific and highly effective medicine, the treatment of COVID-19 patient is very challenging, and neutralizing antibodies have great potential for treatment. There are many neutralization antibodies have been reported, most of them are targeting RBD, especially RBM, some are targeting NTD and S2. However, antibodies that directly against the RBM region may induce escape mutations. Therefore, more neutralization antibodies are still needed, especially antibodies that targeting conserved and functionally important regions on S protein, theoretically, which can reduce the occurrence of mutations and may exert a broad-spectrum antiviral effect against coronaviruses.

In this article, neutralizing antibody FD20 was isolated from COVID-19 convalescent patients through a phage display strategy. FD20 targets a conservative position and have a broad-spectrum antiviral effect. For characterization and functional study, the authors demonstrated FD20's high affinity to ACE2, the effectiveness on cell-based neutralization assays. In order to understand the mode-of-action, the authors resolved the co-crystal structure of FD20-S protein. Overall, the aim and main findings of this study are important, the design and logic are clear, the conclusions could be supported by the results. The manuscript is well-written. Thus, this reviewer endorses the publication of this work after revision.

Major points :

1. In most cases, the destination of a neutralization is for therapeutics. Thus, the in vivo (animal model) validation of its efficacy for protection or treatment is the key. Unfortunately, for animal model experiments, only marginal efficacy was observed. The authors need to improve the efficacy through practical ways, at least, the authors need to thoroughly discuss this.
2. The in vitro affinity of FD20 to ACE2 is very high, while the in vivo efficacy is very low. A reasonable explanation is necessary? Is it possible that the in vivo stability of FD20 is low? Or FD20 was eliminated rapidly through some yet-to-be-discovered mechanism? The reviewer suggests the authors to test the distribution of FD20 in animal model at different time points.
3. For kinetic measurement, to assure the reliability of the determined Kd, the range of the serial dilution should cover the Kd. However, for Fig. 1B, 1D and 1E, this is not the case.

Specific points:

1. For COVID-19, both the pandemic (especially new mutants) and the research are changing very fast. Please update the related information in the revision. For example, the mutant that cause the surge of COVID-19 cases in India need to be discussed.
2. Except RBD, neutralization antibodies targeting NTD and S2 have also been identified. These should also be introduced.
3. Fig. 1B, D and E, concentration information are missing.
4. Line 52 "perhaps all coronavirusee...", how can the author claim "all"?
5. Line 149, what are the "RBM-mAbs" refer to?
6. Line 189, the injection volume needs to be calculated based on the weight of the hamster.
7. Is there a repetition of Fig. 2A to C? Please clarify.
8. It's not clear how the experiment was performed for Fig. 4D, please clarify.
9. Format
 - (1) Page 5, line 138, pseudovirus particles (pp), the abbreviation has already given on Page 4, line 101, it does not need to explain again.
 - (2) Page 11, line 270, "Starr et al", should be "Starr et al."

Referee #3 (Remarks for Author):

Uncovering a conserved vulnerability site in SARS-CoV-2 by a human antibody

Authors: Tingting Li Hongmin Cai Yapei Zhao Yanfang Li Yanling Lai Hebang Yao Liu Liu Zhou Sun Martje Fentener van Vlissingen Thijs Kuiken Corine H. GeurtsvanKessel Ning Zhang Bingjie Zhou Lu Lu Yuhuan Gong 5 Wenming Qin Moumita Mondal Bowen Duan Shiqi Xu Audrey S Richard Hervé Raoul Jianfeng Chen Chenqi Xu Ligang Wu Haisheng Zhou Zhong Huang Xuechao Zhang , Jun Li , Yanyan Wang Yuhai Bi Barry Rockx Junfang Chen, Feilong Meng, Dimitri Lavillette, Dianfan Li

Li and colleagues present a paper describing their efforts to isolate and characterize antibodies using a phage display library created from a SARS-CoV-2 (SARS2) convalescent patient. This effort led to the discovery of an antibody designated as FD20. Using functional and structural analyses they show that the antibody binds to the receptor binding domain (RBD) in a region that is outside of the ACE2 binding site (termed receptor binding motif or RBM in the manuscript). This antibody also has some cross-reactivity with SARS-CoV-1 (SARS1) and the binding location does not yet appear to be a mutational hot-spot within the RBD domain. The authors make 8 major claims in their report that at times are over-reaching in their conclusion or significance that would need to be addressed or modified.

A. The manuscript makes the following claims:

1. FD20 targets an uncharacterized epitope in RBD
 - mAb competition assays should be performed using antibodies with known epitopes to determine if the epitope is unique.
2. FD20 has ultra-high affinity (22pM)

Comments:

- The best practice for defining antibody affinity is to use titrate Fab or scFv (i.e., single binding arm) with a surface bound antigen. This minimizes avidity effects and allows for 1:1 fitting models to be applied to the acquire model. However, the claim of ultra-high affinity for FD20 is made using the full IgG molecule which is subject to avidity, which is not appropriate. Instead the appropriate affinity mentioned for the abstract and major claims should be 2.1 nM (Figure 1B).
 - Even if IgG is used for affinity, while the affinity of 22pM is high (Figure 1D), it is not unusual to have IgG affinities in the pM or sub-pM range.
 - The IgG affinity measurements (Figure 1D/E) appear to have been calculated using a 1:1 fitting model. Since it is a bivalent interaction, the bivalent model should instead be applied. This result should have a k_{1on} , k_{2on} , k_{1off} and k_{2off} rates that is a measure of the avidity.
 - The methods do not indicate whether a global or individual curve fitting was performed for Figure 1B/D/E. This can impact the interpretation of the results.
 - In Figure 1B/D/E only the fitted curves are provided. The raw data curves should be shown in the same graphs.
3. FD20 is a potent neutralizer of B.1.1.7 and B.1.351
 - The neutralization potency of the wt, B.1.1.7 and B.1.351 was stable but poor (i.e., 1.6 to 3.2 ug/mL) relative to other antibodies which are at least 100-1000-fold lower. Therefore, it is not appropriate to refer to FD20 as potent against any SARS-CoV-2 virus or variant.
 - Figure 2D shows various antibody neutralization IC50 and fold-change relative to wt neutralization for that antibody. The table is colored in heat-map fashion relative to the fold change for the antibody on a scale that starts from 0 to >1500. This results in a pink color for those that are 1-fold change or less and darker red color for those higher than 1-fold. This color scheme is misleading for several reasons. First, pink coloration for no change or increased potency could incorrectly suggest to the reader that potency is in fact worse. Second and more importantly, it does not account for variable initial potencies of the comparator antibodies. Therefore, antibodies that started with lower IC50 values than FD20 appear to be worse than FD20 when their IC50

is equivalent to it. For example, 10933 IC50 for B.1.351 is ~10-fold higher than wt but its IC50 is lower than FD20. The color coding misleads the reader into thinking it is worse than FD20.

- A minor comment on the table is that darker red colors make it difficult to read the black text. I would suggest using a white font for the darker colored boxes.

4. FD20 is modestly cross-reactive with SARS1

- This claim is supported by the results

5. FD20 reduces viral replication in hamsters

- While the lung titer PCR is statistically significantly lower, it is less than 10-fold lower and there is no difference in Lung TCID50 and Nasal titer PCR or TCID50 (Figure 3). Stating the there viral replication is lower is over-reaching when considering the totality of the data.

6. FD20 binds to a site that that is ideal for low escape emergence

- While using previously published data supports this idea, the authors should validate the critical contacts in the epitope using mutagenesis and assays for binding and neutralization

- There are millions of sequences in the GSAID databased available for comparison. Therefore, only looking at 2-50 sequences (Figure 4E and text) is a vast under sampling. The authors should take advantage of websites like cov.lanl.gov or others that can rapidly look at the residue in question and determine the true level of conservation.

- Furthermore, the propensity for susceptibility to escape mutation was not formally tested.

7. FD20 neutralizes by destroying the spike

- Prolonged incubations and fitting of models were used as evidence that FD20 destroys S. However, modeling is not proof. To more definitively show this they should perform EM analysis with immediately made grids and progressively increasing time

8. The "results reveal a conserved vulnerability site in SARS-CoV-2 Spike and perhaps all coronaviruses for the development of potential antiviral drugs."

- The statement is overreaching and should be toned down.

- See claim 6 above discussing conservation in SARS2. There should be more comparison for SARS1 and other coronaviruses

B. In the virologic entry assays and order of addition experiment presented in Figures 5&6, significance claims are made for changes in FD20 responses that are statistically significant but likely not virologically significant. In entry assays, virologically significant changes are at least 5-fold but ideally 10-fold. Therefore, claims about order of addition, etc need to be tempered.

Minor:

- Line 71: The distinction between the receptor binding domain and the receptor binding motif (RBM) may be confusing for some readers. It may be better to simply refer to the motif as the ACE2 binding site instead.

- Lines starting at 68 and remainder of paragraph 2 of the introduction: The manuscript uses a very simplified structural classification of RBD binding antibodies. Their classification scheme breaks up binding into those within the ACE2 binding site and those that bind outside the ACE2 binding site. However, this classification schema does not account for the known ability of the SARS2 RBD to be in two structural positions, the down (a.k.a. closed) or up (a.k.a. open) positions^{1,2}. This is an important distinction because ACE2 can only engage with RBD is in the up position and some epitopes are partially or fully obscured with RBS is in the down position^{1,2}. Thus, the field has generally chosen to use the Barnes classification³ that accounts for RBD position and antibody binding to the ACE2 site. This section and the manuscript would be improved if the Barnes classification was used.

- Line 78: This reviewer needed to look up the definition of what the "core regions of the RBD" is. I would suspect others are not aware. Please define for the reader what this is.

- Line 79: There are many antibodies that bind outside the ACE2 binding domain (Barnes Class III and IV) that are not cross-reactive, so it is probably too strong to say "non-RBM mAbs are usually cross-reactive". I would suggest changing usually to often.

- Line 124: Recommend avoiding terms like remarkable increase in activity rather say with that increase was.

- Line 124-126: The improved potency from 119.4 to 11.9 nM is 10-fold better but this is not surprising given the increased avidity expected in a full IgG vs an scFv. Thus, it is not too remarkable a finding.

- Line 139: did you mean to say 1.9-fold increase instead of 1-fold increase.

- Lines 321-325. Why is the IC50 in the cell fusion assay so much different than the neut assays? Please define what you mean by "effective"?

- Lines 340-344: Please provide or reference the data supporting the claim that RBM-mAbs have tight binding and slow dissociation.

- Line 391: how is synergy determined

- Lines 398-411.

- o It appears that S-2P protein was used. This should be explicitly used instead of S whenever appropriate.

- Throughout the manuscript there are word choices and sentence structure formats that cause the reader to have to infer or guess what is meant. Also, there are uses of words that are not usual, like iconic, that are not typical for scientific writing and may mislead the reader to either think the writers are making claims that I do not think they mean to. Another example on Line 256 uses the phrase "could inhibit" sounds like a hypothesis, but I think it is a statement. Therefore, the manuscript would benefit from being reviewed and edited by a professional English language scientific editor or someone similar who can provide the words and sentences that convey the authors meaning with more clarity.

References

1. Wrapp, D. et al. Cryo-EM structure of the 2019-nCoV spike in the prefusion conformation. *Science* 367, 1260-1263 (2020).
2. Hsieh, C. et al. Structure-based design of prefusion-stabilized SARS-CoV-2 spikes. *Science* 369, 1501-1505 (2020).
3. Barnes, C. O. et al. SARS-CoV-2 neutralizing antibody structures inform therapeutic strategies. *Nature* 588, 682-687 (2020).

Responses to Reviewer Comments

Referee #1

The manuscript by T. Li et. al. reports isolation and characterization of a non-RBM monoclonal antibody (FD20) from COVID-19 convalescent patients with good neutralization activity against SARS-CoV-2. Although many neutralized Abs for SAR-CoV-2 have been reported before, the importance of FD20 is that it targets to a conserved epitope and neutralizes SARS-CoV-2 including the current concerning lineages B.1.1.7 and B.1.351. The authors have done experiments from different angels to prove the activities and potential mechanisms. I only have some minor concerns:

We thank the reviewer for the positive comments and specific suggestions to improve our manuscript.

1. Fig. 1A: For the screening, the authors mentioned that B cells from three convalescent patients were pooled. Are there any criteria for choosing the convalescent patients? At the end of the screening, how many neutralized antibodies were obtained besides scFD20?

We performed a quick ELISA assay using RBD and chose three patients based on the ELISA results.

We obtained a total of 52 RBD-binding antibodies (as judged by their ability to shift gel-filtration peak of RBD), seven of which (including scFD20) showed >50% neutralization at 1 μ M concentration. The information has been included in the revised manuscript (**Appendix Fig. S1**) and the sequences of the seven scFv antibodies have been included in **Appendix Table S1**. For your information, **Appendix Fig. S1** is attached below.

Appendix Figure S1. Identification of neutralizing antibodies. (A) Fifty-three scFv antibodies that bind to RBD on gel filtration were subjected to neutralization assays using SARS-CoV-2 pp and scFv at indicated concentrations.

2. Line 184 We next investigated the in vivo potential of FD20 by intraperitoneal injection 6 h before intranasal inoculation of hamsters by live viruses (Figure 3): Is this sentence meaning that FD20 was injected first and then virus infection after 6h? It looks strange since normally for antibody treatment, the animals should be infected with virus first and then inject antibody for treatment.

The reviewer is correct that the hamsters were treated with the FD20 6 hours prior to intranasal challenge with SARS-CoV-2. Virus-neutralizing antibodies can be used prophylactically to prevent infection in high-risk cases, such as vulnerable individuals with underlying medical conditions which may not be vaccinated, health care providers, and individuals with exposure to confirmed cases of COVID-19.

We decided to use this experimental design as a proof-of-principle to determine the maximal effect of FD20. Because of the incomplete protection of FD20 and the limited available BLS3 slots during the ongoing SARS-CoV-2 crisis, we were unable to secure timeslots to test the efficacy of treatment at time points after initiation of infection to determine the therapeutic window.

3. Fig. 3B and 3E: The virus titers in either the lung or the nasal tissues determined by PCR and TCID₅₀ were not consistent. For example, there are two lung tissues showing 0 TCID₅₀ (Fig. 3C) and the PCR results (Fig. 3B) are quite high. What could be the reasons behind? It is quite hard to understand why the titers were varied so much between the PCR method and the TCID₅₀ method.

In general, some variability in viral titers/pathology etc. is to be expected, i.e., as no fully inbred strain of hamsters is available. Detection of SARS-CoV-2 is most sensitive and reproducible when using qRT-PCR, which is the data we base our conclusions on. However, in the context of potential for transmission, we also performed virus titrations on homogenates of lung and nasal turbinates, to detect the presence of infectious virus.

Consistently, the animals with the lowest RNA levels were also the animals in which infectious virus was not detected.

Unfortunately, the TCID₅₀ method is less sensitive and due to regular toxicity of these homogenates in cell culture increases the limit of detection up to 1000 TCID₅₀/mL. Therefore, while we could not detect infectious virus, in theory, there could still be up to 1000 TCID₅₀ present in tissues. In addition to the issue described above, viral titers in nasal turbinates are even more variable due to the fact that we normalize all titers to grams of tissue. For lung tissue, we can typically take samples ~100 mg, which then only requires us to multiply by 10. However, in the case of nasal turbinates, 10 mg or less of tissue is typically harvested and thus increasing variability when normalizing to g tissue.

Referee #2

COVID-19, the pandemic caused by SARS-CoV-2, is still threatening thousands of millions of people around the world. In the absence of specific and highly effective medicine, the treatment of COVID-19 patient is very challenging, and neutralizing antibodies have great potential for treatment. There are many neutralization antibodies have been reported, most of them are targeting RBD, especially RBM, some are targeting NTD and S2. However, antibodies that directly against the RBM region may induce escape mutations. Therefore, more neutralization antibodies are still needed, especially antibodies that targeting conserved and functionally important regions on S protein, theoretically, which can reduce the occurrence of mutations and may exert a broad-spectrum antiviral effect against coronaviruses.

In this article, neutralizing antibody FD20 was isolated from COVID-19 convalescent patients through a phage display strategy. FD20 targets a conservative position and have a broad-spectrum antiviral effect. For characterization and functional study, the authors demonstrated FD20's high affinity to ACE2, the effectiveness on cell-based neutralization assays. In order to understand the mode-of-action, the authors resolved the co-crystal structure of FD20-S protein. Overall, the aim and main findings of this study are important, the design and logic are clear, the conclusions could be supported by the results. The manuscript is well-written. Thus, this reviewer endorses the publication of this work after revision.

We thank the reviewer for the complimentary remarks and specific comments below to improve our work.

Major points:

1. In most cases, the destination of a neutralization is for therapeutics. Thus, the *in vivo* (animal model) validation of its efficacy for protection or treatment is the key. Unfortunately, for animal model experiments, only marginal efficacy was observed. The authors need to improve the efficacy through practical ways, at least, the authors need to thoroughly discuss this.

We agree with the reviewer on this matter. We have tried to increase the binding affinity of FD20 by structure-based mutagenesis, an approach we have successfully used in our previous work with an RBD-targeting nanobody (*Nat Commun* 2021 **12**:4635). To this end, we designed 6 mutants (**Fig R1 below**) and compared their neutralizing activity with FD20. Unfortunately, the effort did not yield mutants with improved neutralizing activity (**Fig R1**). In the future, *in vitro* maturation techniques such as soft-randomization could be employed to obtain tighter mutants. Alternatively, FD20 may be fused to high-affinity mAbs or nanobodies to seek higher potency.

Figure R1. Structure-based design and neutralization results of mutants that are intended for improved neutralizing activity. (A-F) Structural illustration of mutations intended to improve binding affinity by introducing potential salt-bridges (A, D, and E) or by strengthening hydrophobic interactions (B, C, and F). In the expanded view, the up panel shows the wild-type FD20 residue and the bottom panel shows the residue after mutation. Note the structure figure at the bottom was generated by ‘virtual mutation’, i.e., not by experiments. RBD residues are shown as green and labeled as grey text. (G) Neutralization assay of the wild-type FD20 and the mutants using antibody concentrations at 100 nM and 10 nM. *L*- and *H*- indicates light and heavy chains, respectively.

A paragraph has been added to the Discussion as follows:

Since the outbreak begins, thousands of neutralizing antibodies against SARS-CoV-2 have been reported (<http://opig.stats.ox.ac.uk/webapps/coronavirus>) and hundreds have their epitopes

structurally characterized, most of which targets RBM and displays IC₅₀ values 1-2 orders of magnitude^{16,17,60} lower than FD20. For functional reasons, RBM is more exposed than other regions. Thus, the higher abundance of mAbs targeting the RBM is perhaps not unexpected. It is also possible that the non-RBM-targeting mAbs are generally less potent than RBM-targeting mAbs, and are thus less prioritized for study and less reported. Notably, the IC₅₀ of FD20 (1.7 µg/mL) is similar to most of non-RBM-targeting mAbs (S2A4^{ref.61}, 3.5 µg/mL; H014^{ref. 62}, 5.4 µg/mL; EY6A²³, 0.07 and 20 µg/mL in two sets of experiments; CR3022^{ref. 63}, 5.2 µg/mL for SARS-CoV; CR3022-P384A^{ref. 63}, 3.2 µg/mL for SARS-CoV-2), although more potent ones have also been reported (S309^{ref. 21}, 0.08 µg/mL; COVA1-16^{ref. 64}, 0.02 µg/mL). Another possible reason for the relatively low neutralizing activity is its modest binding affinity (K_D of 5.6 nM for the scFv form) for RBD. The structural information may be used to rationally design gain-of-function mutants to increase binding affinity and hence neutralizing activity, as demonstrated in our previous study with a neutralizing nanobody⁶⁵. *In vitro* maturation methods such as soft randomization⁶⁶ and based-editor guided *Ex vivo* maturation⁶⁷ may also be employed to increase affinity. Alternatively, because the unusual FD20 epitope does not overlap with most mAbs and nanobodies, fusing FD20 with high-affinity nanobodies may increase the apparent binding affinity by avidity effects⁴⁷.

The *in vivo* protection of FD20 was also modest. This could be due to the relatively low neutralization activity *in vitro*. Alternatively, it may be related to half-life which is not tested in this study. Given that hamsters are becoming a common animal model for SARS-CoV-2 infection, and that little *in vivo* stability information is available for human mAb in hamsters, this issue warrants future investigation.

2. The *in vitro* affinity of FD20 to ACE2 is very high, while the *in vivo* efficacy is very low. A reasonable explanation is necessary? Is it possible that the *in vivo* stability of FD20 is low? Or FD20 was eliminated rapidly through some yet-to-be-discovered mechanism? The reviewer suggests the authors to test the distribution of FD20 in animal model at different time points.

As the reviewer #3 pointed out, the previously stated high affinity (22 pM) between the IgG FD20 and RBD is probably due to avidity effects. Instead, the affinity between the scFv version and RBD may be more appropriate, which is not very high (5.6 nM in the revised manuscript). Therefore, the low *in vivo* efficacy may be related to the modest affinity.

We considered the reviewer's comments about measuring the *in vivo* stability. However, our institute (Shanghai Institute of Biochemistry and Cell Biology or Institut Pasteur of Shanghai) does not have animal housing and protocols for Syrian hamsters. Therefore, regretfully, we did not do the experiments. Instead, we have added a paragraph to discuss this, as shown in the responses to the previous point.

3. For kinetic measurement, to assure the reliability of the determined K_d, the range of the serial

dilution should cover the K_D . However, for Fig. 1B, 1D and 1E, this is not the case.

We thank the reviewer on this point. We have repeated the binding kinetics using lower concentrations until the BLI signal was too low to be reliable. We should note that the binding kinetics are now slightly different than previously reported. For your information, the new results are attached below (Fig 1B/C).

Fig. 1. Isolation and characterization of scFD20. ... (B, C) Kinetics of the binding between the scFv of FD20 (scFD20) and RBD from SARS-CoV-2 (B) or SARS-CoV (C). Biolayer interferometry (BLI) assay was performed with RBD immobilized and scFD20 as analyte at indicated concentrations (nM). Raw data are shown in color and fitted lines are shown in grey. K_D values were calculated using 1:1 global fitting.

Specific points:

1. For COVID-19, both the pandemic (especially new mutants) and the research are changing very fast. Please update the related information in the revision. For example, the mutant that cause the surge of COVID-19 cases in India need to be discussed.

We are trying to follow the development of Variants of Concern and we generated S derived from most of them. We have performed neutralization assays using SARS-CoV-2 Delta pp, as well as the Gamma pp. The results showed that FD20 can neutralize both variants with the same potency against the wild-type SARS-CoV-2 (1.0 fold for Gamma and 1.1 fold for Delta).

The results have been updated in **Fig. 2D**.

2. Except RBD, neutralization antibodies targeting NTD and S2 have also been identified. These should also be introduced.

Done.

3. Fig. 1B, D and E, concentration information are missing.

The following note has been added to the caption:

Biolayer interferometry (BLI) assay was performed with RBD immobilized and scFD20 as analyte at indicated concentrations (nM).

4. Line 52 "perhaps all coronaviruse...", how can the author claim "all"?

This overstatement has been removed in the revised manuscript.

5. Line 149, what are the "RBM-mAbs" refer to?

This inaccurate expression has been changed to "RBM-targeting" mAbs throughout the manuscript.

6. Line 189, the injection volume needs to be calculated based on the weight of the hamster.

In rodents, given the number of animals, typically the same amount of mAb is given assuming similar weights. We have revised the text to include the following:

...as well as in viral RNA in the lungs of hamsters treated with FD20 (5 mg doses; 62-75mg/kg).

7. Is there a repetition of Fig. 2A to C? Please clarify.

Sorry for the confusion. There is no repetition.

Fig. 2A is for pseudoviruses and **Fig. 2C** is for authentic viruses. We have modified the legends to clarify this point, as follows.

... (A) Neutralization assay of FD20 using SARS-CoV-2 pp harboring spike derived from four Variants of Concern as indicated.

...(C) The plaque-reduction neutralization assay using two authentic SARS-CoV-2 strains: the Germany isolate (BetaCoV/Munich/BavPat1/2020) and UK/B1.1.7.

8. It's not clear how the experiment was performed for Fig. 4D, please clarify.

The legends of **Fig 4D** has been modified to clarify the experimental design:

(D) Modest perturbation of scFD20 for RBD-ACE2 interaction. A sensor with RBD immobilized was soaked in 200 nM of scFD20 before being further soaked in scFD20-containing buffer with (red) or without (blue) 25 nM of ACE2 for BLI signal recording. As a control, the ACE2-RBD interaction was monitored in the absence of scFD20 (black).

9. Format

(1) Page 5, line 138, pseudovirus particles (pp), the abbreviation has already given on Page 4, line 101, it does not need to explain again.

We thank the reviewer for the careful reading.

(2) Page 11, line 270, "Starr et al", should be "Starr et al."

Done.

Referee #3

Li and colleagues present a paper describing their efforts to isolate and characterize antibodies using a phage display library created from a SARS-CoV-2 (SARS2) convalescent patient. This effort led to the discovery of an antibody designated as FD20. Using functional and structural analyses they show that the antibody binds to the receptor binding domain (RBD) in a region that is outside of the ACE2 binding site (termed receptor binding motif or RBM in the manuscript). This antibody also has some cross-reactivity with SARS-CoV-1 (SARS1) and the binding location does not yet appear to be a mutational hot-spot within the RBD domain. The authors make 8 major claims in their report that at times are over-reaching in their conclusion or significance that would need to be addressed or modified.

We thank the reviewer for the careful evaluation of our work, and constructive comments and specific suggestions below to avoid overstatements.

A. The manuscript makes the following claims:

1. FD20 targets an uncharacterized epitope in RBD
 - mAb competition assays should be performed using antibodies with known epitopes to determine if the epitope is unique.

When we first submitted our manuscript to other journals in September last year, we did competitive binding with the then available mAbs (REGN10933, REGN10987, CV30, CB6, CR3022) and we found that FD20 bound to RBD in the presence of other mAbs. Because the number of mAbs in the literature has grown very rapidly since, producing all mAbs with known epitopes (over 200 now) for competitive binding assays has become impractical. We have therefore removed this claim. The revised manuscript now focuses on the conservation of the epitope.

2. FD20 has ultra-high affinity (22pM)

Comments:

- The best practice for defining antibody affinity is to use titrate Fab or scFv (i.e., single binding arm) with a surface bound antigen. This minimizes avidity effects and allows for 1:1 fitting models to be applied to the acquire model. However, the claim of ultra-high affinity for FD20 is made using the full IgG molecule which is subject to avidity, which is not appropriate. Instead the appropriate affinity mentioned for the abstract and major claims should be 2.1 nM (Figure 1B).

We now use the K_D for the scFv only in the revised manuscript. Note that we have repeated the K_D measurement using lower concentration as suggested by Reviewer #2, and the K_D is now reported as 5.6 nM.

- Even if IgG is used for affinity, while the affinity of 22pM is high (Figure 1D), it is not unusual to have IgG affinities in the pM or sub-pM range.

We have removed the results of the apparent binding kinetics information for IgG-RBD.

- The IgG affinity measurements (Figure 1D/E) appear to have been calculated using a 1:1 fitting model. Since it is a bivalent interaction, the bivalent model should instead be applied. This result should have a k_{1on} , k_{2on} , k_{1off} and k_{2off} rates that is a measure of the avidity.

We have repeated the experiment using low concentrations of analyte as Reviewer #2 suggested and processed the data with both 1:1 and 1:2 fitting. The results (**Fig R2** below) showed very low k_{2on} . In addition, the 1:1 fitting matches the raw data better with a higher R^2 than the 1:2 fitting. The results may suggest low coating density and the minor existence of bridged complexes, although we do not have further evidence for this.

As the reviewer points out, the use of K_D for scFv-RBD is more appropriate. Therefore, we have removed the results for the binding between IgG and RBD in the revised manuscript.

Fig R2. BLI assay and the two different fitting mode for the binding between IgG FD20 and RBD. (A) SARS-CoV-2 RBD and FD20, 1:1 fitting; (B) SARS-CoV RBD and FD20, 1:1 fitting; (C) SARS-CoV-2 RBD and FD20, 1:2 fitting; (D) SARS-CoV RBD and FD20, 1:2 fitting. (E) Summary of the binding kinetic parameters by the two fitting methods.

- The methods do not indicate whether a global or individual curve fitting was performed for

Figure 1B/D/E. This can impact the interpretation of the results.

We thank the reviewer for this comment. Global curve fitting was used for all binding kinetics. The information has been added in the Methods.

- In Figure 1B/D/E only the fitted curves are provided. The raw data curves should be shown in the same graphs.

Both the raw data and the fitted curve are now included in **Fig 1B/C**.

3. FD20 is a potent neutralizer of B.1.1.7 and B.1.351

- The neutralization potency of the wt, B.1.1.7 and B.1.351 was stable but poor (i.e., 1.6 to 3.2 ug/mL) relative to other antibodies which are at least 100-1000-fold lower. Therefore, it is not appropriate to refer to FD20 as potent against any SARS-CoV-2 virus or variant.

We have removed the claim that FD20 is a potent neutralizer and we have discussed ways to improve its potency in the Discussion.

- Figure 2D shows various antibody neutralization IC₅₀ and fold-change relative to wt neutralization for that antibody. The table is colored in heat-map fashion relative to the fold change for the antibody on a scale that starts from 0 to >1500. This results in a pink color for those that are 1-fold change or less and darker red color for those higher than 1-fold. This color scheme is misleading for several reasons. First, pink coloration for no change or increased potency could incorrectly suggest to the reader that potency is in fact worse. Second and more importantly, it does not account for variable initial potencies of the comparator antibodies. Therefore, antibodies that started with lower IC₅₀ values than FD20 appear to be worse than FD20 when their IC₅₀ is equivalent to it. For example, 10933 IC₅₀ for B.1.351 is ~10-fold higher than wt but its IC₅₀ is lower than FD20. The color coding misleads the reader into thinking it is worse than FD20.

We have modified the color coding to avoid misleading.

We now use cold colors for increased potency.

For the second issue, we now provide two heat maps: one for the fold change and one for the IC₅₀ values to let readers judge with her/his criteria.

- A minor comment on the table is that darker red colors make it difficult to read the black text. I would suggest using a white font for the darker colored boxes.

We have removed the dark red color and the contrast is now more obvious. We thank the reviewer for the useful suggestion.

4. FD20 is modestly cross-reactive with SARS1

- This claim is supported by the results

OK.

5. FD20 reduces viral replication in hamsters

- While the lung titer PCR is statistically significantly lower, it is less than 10-fold lower and there is no difference in Lung TCID50 and Nasal titer PCR or TCID50 (Figure 3). Stating the there viral replication is lower is over-reaching when considering the totality of the data.

We agree with the reviewer that only a modest protective effect in the lung is observed in FD20-treated animals. However, the significant reduction in viral RNA is supported by the significant reduction in viral antigen in the lungs. Although there was no significant difference in infectious titers, this is likely due to the large variation in viral titers, as 4 out of 5 animals had lower titers compared to PBS controls.

We have revised the text in the manuscript as follows:

...as well as a small but significant reduction in viral RNA in the lungs of hamsters treated with FD20 (5 mg doses) compared to those that were untreated ($p = 0.005$) (Figure 3B).

6. FD20 binds to a site that that is ideal for low escape emergence

- While using previously published data supports this idea, the authors should validate the critical contacts in the epitope using mutagenesis and assays for binding and neutralization

We have made 15 mutations for the 12 epitope residues and compared the binding kinetics and neutralizing potencies of the mutants with that for the wild-type (**Table 2**). The results showed that 1) three mutants lost infectivity; 2) the binding affinity was variably affected by the mutations but the neutralization activity was not much affected; the two most dramatic mutants D426G and K462A had a ~2-fold IC_{50} compared to that of the wild-type.

For your information, the results are attached below:

To further test whether the naturally occurring mutations affect FD20's binding and neutralizing activity, we have designed a panel of mutants as listed in **Table 2** using the following criteria. For each epitope residue, the most frequent mutation was chosen unless the substitution was conservative, in which case amino-acid with the most dramatic changes was selected. For example, Arg466 was mutated to isoleucine instead of lysine. In addition, the charge of Lys462 and Glu465 were either eliminated by alanine mutation (not naturally occurring) or reversed by the mutations K462E and E465K (found in the database with low frequencies) (**Fig EV1B**), respectively; such mutations were expected to weaken the FD20-RBD binding because both residues provided intermolecular salt bridges (**Fig 4B**). Finally, the epitope-proximal residue Asp467 was mutated to tyrosine to test its structural/functional

constraint. The mutations were introduced to RBD for binding assay separately with ACE2 (**Fig EV2A**) and FD20 (**Fig. EV2B**), and to S protein in SARS-CoV-2 pp for infectivity (**Fig EV2C**) and neutralization assay (**Fig EV2D**). In addition, the expression level and processing of S (whether S is cleaved to produce S1 and S2) were assessed by Western blotting as quality control for trafficking and maturation (**Fig EV2E, EV2F**).

Among the 15 tested RBD mutants, W353R, R355T, and D467Y showed no expression in insect cells (**Table 2**). Although the corresponding full-length S mutants were detected in mammal cells, they were not processed effectively as evidenced by the lack of the S1 subunit on SDS-PAGE (**Fig EV2E, EV2F**). These results jointly suggest the mutants were misfolded. As expected, mutations of Lys462 and Glu465 affected FD20-binding. However, the mutations either reduced infectivity drastically (E465K, 5% of the wild-type), or remained sensitive to FD20, displaying similar IC₅₀ values (**Fig. EV2D, Table 2**). Similarly, mutation of Arg466, another residue that contributes a salt-bridge, also impaired binding but had little effect on the neutralization (**Table 2**). Finally, the rest of the mutation did not weaken the binding activity, and the neutralizing activities of FD20 for these mutants also did not change noticeably (**Table 2**). The two most escaping mutants were D428G and K462A, which showed a ~2-fold IC₅₀ compared to the wild-type SARS-CoV-2 pp. Taken together, the analysis showed that the FD20 epitope is highly conserved, and the experimental results confirmed that some are indeed highly structurally (Trp353/Asp467/Arg355/Glu465) constrained as suggested by Starr *et al.*^{38,54}; in the case of mutants with similar infectivity to the wild-type, FD20 remained effective for neutralization, indicating FD20's potential to resist escape mutants.

Table 2. Summary of biophysical and biological characteristics of mutants at the FD20 epitope.

Constructs ^a	Mutation frequency (per million) ^b	ACE2-binding ^c	FD20-Binding ^c	S Expression ^e	S Processing ^f	Infectivity (% of wt)	IC ₅₀ , nM [fold]
Wild-type	n/a	++	++	++	++	100	12.0 [1.0]
W353R	2.7 [4.1]	n.d. ^d	n.d. ^d	++	-	0.7	n.d. ^d
N354K	172.0 [391.7]	+++	++	++	++/-	85.6	11.9 [1.0]
R355T	0.7 [0.7]	n.d. ^d	n.d. ^d	++	-	0.6	n.d. ^c
P426S	14.3 [16.3]	+++	++	++	+	55.7	12.7 [1.1]
D428G	15.0 [29.3]	++	++	++	++/-	72.1	23.7 [2.0]
L461I	8.8 [10.2]	+++	++	++	++	92.6	14.8 [1.2]
K462A	0 [21.1]	++	+	++/-	++	119.0	25.2 [2.1]
K462E	5.1 [21.1]	+++	+/-	++	++	67.9	16.6 [1.4]
P463S	57.2 [62.0]	++	++	+	+/-	40.5	12.4 [1.0]
F464S	1.4 [9.5]	+++	++	+	+/-	26.0	15.9 [1.3]
E465A	0 [21.1]	+++	+/-	+	++/-	38.3	16.7 [1.4]
E465K	0.7 [21.1]	+++	+/-	+	-	5.0	n.d. ^d
R466A	0 [10.2]	++/-	++/-	+	+/-	21.4	17.7 [1.5]
R466I	2.7 [10.2]	++	++/-	+	+	77.1	15.0 [1.3]
D467Y	1.4 [4.1]	n.d. ^d	n.d. ^d	++	-	0.8	n.d. ^d

^aBinding was performed with the wild-type or mutations on RBD, and infectivity and neutralization assays were performed with pseudoviruses harboring the wild-type or mutant S protein. ^bFrequency of the particular mutation followed by mutation frequency at this residue in brackets as of June 9, 2021 (www.gisaid.org; cov.lanl.gov). ‘0’, no such mutation has been reported. ^dBinding properties were evaluated by the biolayer interferometry (BLI) signal and apparent binding affinity. ^cNot determined because either the construct had no expression in insect cells (for binding) or the mutant SARS-CoV-2 pp had low infectivity (for neutralizing assay). ^eExpression level of S was evaluated by Western blotting using an anti-S1 antibody. ^fProper processing and trafficking were evaluated by Western blot of S1 in the cell culture supernatant. Binding kinetics and expression level are rated from high to low following the sequence of “+++”, “++/-”, “+”, “+/-”, and “-”. The data for binding (BLI curve), expression (Western blot), infectivity, and neutralization are in **Fig. EV2**.

- There are millions of sequences in the GSAID databased available for comparison. Therefore, only looking at 2-50 sequences (Figure 4E and text) is a vast under sampling. The authors should take advantage of websites like cov.lanl.gov or others that can rapidly look at the residue in question and determine the true level of conservation.

We thank the reviewer for pointing out this very useful resource to us. We have updated the

information in **Fig. EV1**. For your information, we have attached the relevant texts below:

...Indeed, the mutation frequencies of the FD20 epitope residues from ~1.5 million deposited sequences⁵³ (cov.lanl.gov) were relatively low. Thus, there is only one mutation reported for Arg355 (R355T) (as of June 9th, 2021), and the mutation frequencies (per million) of the rest residue were below 70, with the exception for Asn354 which was 391 (**Fig EV1**).

^aFrom a total of ~1.47 million sequences as of June 9, 2021 (www.gisaid.org; cov.lanl.gov). ^bNear the FD20 epitope.

Figure EV1. Uneven distribution of naturally occurring mutants in RBD. (A) The distribution of naturally occurring mutants in RBD viewed at different angles. Residues are color-coded according to their mutation frequency using the color scheme on the right. The FD20 epitopes are labeled as C α spheres which are color-coded based on their interaction type with FD20 (by sidechain or by the main chain). (B) Details of the mutations of the FD20 epitope residues.

- Furthermore, the propensity for susceptibility to escape mutation was not formally tested.

We agree with the reviewer that this is an important experiment to do. The experiment would require the use of replication-competent viruses. However, we have been unable to locate

resources for the experiment. We recognize the importance of this and we hope to test this in a follow-up study.

7. FD20 neutralizes by destroying the spike

- Prolonged incubations and fitting of models were used as evidence that FD20 destroys S. However, modeling is not proof. To more definitively show this they should perform EM analysis with immediately made grids and progressively increasing time

The incubation time was, at the time, chosen based on the conditions used in a previous publication for CR3022 (50 min and 3 h, *Cell Host Microbe* 2020 **28**:445). Besides, the fact that the control treatment did not cause S-destruction suggests that the 1-h incubation was unlikely an issue. Still, we think a time-course study would give more information. Therefore, we repeated the experiment and show that the S protein started to lose integrity within 3-15 minutes upon FD20 incubation (**Fig. EV5**). This information has been added to the revised manuscript, and as below:

Figure EV5. FD20 destructs S-2P trimer within 3-15 minutes. (A) Negative staining electron microscopy of S-2P (a stable mutant of S, see Methods) without FD20 incubation. (B-D) Negative staining electron microscopy of S-2P immediately after the addition of FD20 (B), or after 3 min (C) or 15 min (D) of FD20 incubation. Bars are indicated in each panel. Two representative images are shown for each treatment.

Regarding the model-fitting:

We intended to convey that the model-fitting analysis was consistent with the experimental

observation that FD20 destructs S. However, the previous writing might have led to the impression that we used model fitting as evidence that FD20 destructs S. We have now re-organized the paragraph to clarify this point (see below).

FD20 neutralizes SARS-CoV-2 by destructing Spike

Aligning the FD20-RBD structure to the full-length Spike structure showed that the FD20 epitope is inaccessible. As shown in **Fig EV4A**, all three FD20 binding sites are buried between the RBD and the N-terminal domain of S1 (NTD) of the adjacent monomers in the ‘closed’ state⁵⁶ when the three RBDs are at a ‘down’ conformation. When aligned to the ‘open’ state with one ‘up’-RBD, the epitope is more exposed (**Fig EV4B**). Nevertheless, notable clashes are still observed between FD20 and NTD including the two glycans linking to Asn165 and Asn234 (**Fig EV4C**)⁵⁶. This raises the possibility that FD20 executes its virucide effect by destructing S, as reported for CR3022 and EY6A^{22,23}.

To test this hypothesis, we performed negative-stain electron microscopy of S-2P (an engineered S mutant containing two stabilizing proline mutations, see Methods) to investigate the impact of FD20 on the S-2P’s integrity. The purified S-2P, but not FD20, showed particles with its typical ‘chicken leg’ shape (**Fig 6C and 6D**) which represents the side-views⁵⁶. This feature was also clearly visible in the 2-D class averages. By sharp contrast, this feature was lost upon FD20 incubation in both the raw images and the 2-D class averages (**Fig 6E**). As a control, S-2P incubated with an unrelated IgG (5E1)⁵⁷ did not show loss of integrity (**Fig 6F**). A time-course experiment showed that the S-2P protein started to lose integrity in the time window of 3-15 minutes upon FD20 incubation (**Fig EV5**). The results suggest that FD20 neutralizes SARS-CoV-2 mainly by destructing S. Because the FD20 epitope is structurally conserved in several other coronaviruses (**Fig 4F**), their corresponding region may be exploited for the development of neutralizing antibodies with a similar mechanism.

8. The "results reveal a conserved vulnerability site in SARS-CoV-2 Spike and perhaps all coronaviruses for the development of potential antiviral drugs."

- The statement is overreaching and should be toned down.

We have changed the statement to “Our results reveal a conserved vulnerability site in the SARS-CoV-2 Spike for the development of potential antiviral drugs”.

- See claim 6 above discussing conservation in SARS2. There should be more comparison for SARS1 and other coronaviruses

We have included the sequence alignment of RBD from SARS-CoV-2, SARS-CoV, MERS-CoV, hCoV-OC43, and hCoV-HKU1 (**Appendix Fig S3C**). These are coronaviruses known to infect humans with an RBD or RBD-like structure. In addition, RaTG13, a coronavirus highly

similar to SARS-CoV-2, is included. The FD20 epitope is conserved between SARS-CoV-2, SARS-CoV, and RaTG13. For the rest of the three, the epitope residues are not conserved owing to the overall low sequence homology (16-21% identity). However, the epitope residues are organized similarly in 3-dimension when structurally aligned (**Fig 4F**).

The following paragraph has been added to the revised manuscript:

...Complete conservation of the epitope residue was observed when comparing to RaTG13^{ref.49}, of which the RBD is 89.5% identical and 94.0% similar to SARS-CoV-2 RBD (**Fig. 4E**). Among five other coronaviruses known to infect humans, three of them (hCoV-OC43, hCoV-HKU, and MERS) contain RBD-like structures⁵⁰⁻⁵². Although the sequence homology at the FD20's epitope was only modest due to their overall low sequence similarity (**Appendix Fig S3C**), the 3-dimensional organization of the epitope region is structurally conserved (**Fig 4F**).

B. In the virologic entry assays and order of addition experiment presented in Figures 5&6, significance claims are made for changes in FD20 responses that are statistically significant but likely not virologically significant. In entry assays, virologically significant changes are at least 5-fold but ideally 10-fold. Therefore, claims about order of addition, etc need to be tempered.

We have modified the text to reflect the small virological changes:

...When the mAbs were added after the *binding* step, an inhibition of entry, although modest in virological context (25%), was detected for FD20...

Minor:

- Line 71: The distinction between the receptor binding domain and the receptor binding motif (RBM) may be confusing for some readers. It may be better to simply refer to the motif as the ACE2 binding site instead.

We thank the reviewer for the suggestion to make the text less confusing but it is more convenient to use RBM in several cases, for example, 'RBM-targeting antibodies' instead of 'ACE2-binding site-targeting antibodies'. Therefore we respectfully ask to keep the term "RBM".

- Lines starting at 68 and remainder of paragraph 2 of the introduction: The manuscript uses a very simplified structural classification of RBD binding antibodies. Their classification scheme breaks up binding into those within the ACE2 binding site and those that bind outside the ACE2 binding site. However, this classification schema does not account for the known ability of the SARS2 RBD to be in two structural positions, the down (a.k.a. closed) or up (a.k.a. open) positions^{1,2}. This is an important distinction because ACE2 can only engage with RBD in the up position and some epitopes are partially or fully obscured with RBS is in the down position^{1,2}. Thus, the field has generally chosen to use the Barnes classification³ that accounts for RBD position and antibody binding to the ACE2 site. This section and the manuscript would be improved if the Barnes classification was used.

Thanks to the reviewer for the suggestion. We have revised the introduction part according to the Barnes classification:

...Current structurally characterized RBD-targeting monoclonal antibodies (mAbs) can be categorized into four classes based on the RBD conformation and their epitope positions relative to the RBM¹⁴. The Class 1 mAbs bind RBM in up-RBD; the Class 2 mAbs target RBM in both up- and down-RBDs; the Class 3 mAbs recognize non-RBM epitopes in up-RBD and the Class 4 mAbs bind to non-RBM epitopes in both up- and down-RBDs. The RBM-targeting mAbs (Class 1 and 2) inhibit viral entry by directly competing with ACE2^{11,13,14,16-18,34}...

- Line 78: This reviewer needed to look up the definition of what the "core regions of the RBD" is. I would suspect others are not aware. Please define for the reader what this is.

The following sentence has been added to the revised Introduction.

“The RBD structure features a core region consisting of 5 β -strands with connecting loops and α -helices, and the RBM which is mainly made of loops and lay on top of the core region ⁹.”

- Line 79: There are many antibodies that bind outside the ACE2 binding domain (Barnes Class III and IV) that are not cross-reactive, so it is probably too strong to say "non-RBM mAbs are usually cross-reactive". I would suggest changing usually to often.

Done with thanks.

- Line 124: Recommend avoiding terms like remarkable increase in activity rather say with that increase was.

We thank the reviewer for this and other advice for scientific writing.

- Line 124-126: The improved potency from 119.4 to 11.9 nM is 10-fold better but this is not surprising given the increased avidity expected in a full IgG vs an scFv. Thus, it is not too remarkable a finding.

We have rephrased this and other similar expressions in the manuscript.

- Line 139: did you mean to say 1.9-fold increase instead of 1-fold increase.

By ‘1-fold’ increase we meant that the increase was 1-fold, resulting in a ~2-fold IC₅₀ value. We made this clearer by rephrasing this as the following:

Notably, FD20 displayed similar neutralizing activity against SARS-CoV-2 pp harboring S mutants from B.1.1.7/P.1/B.1.617.2 and slightly reduced activity against B.1.351 (IC₅₀ value increased to ~2 fold) (**Fig 2A**).

- Lines 321-325. Why is the IC₅₀ in the cell fusion assay so much different than the neut assays? Please define what you mean by "effective"?

We have modified the sentence as the following:

Even at a 10 nM concentration, the cell-cell fusion was effectively suppressed by FD20, displaying a ~60% inhibition.

- Lines 340-344: Please provide or reference the data supporting the claim that RBM-mAbs have tight binding and slow dissociation.

The previous statement on this was incorrect. The binding affinity for the control mAbs are also in the nanomolar range: REGN10933, 3.4 nM; REGN10987, 45.2 nM (*Science* 2020 **369**:1010); CB6, 2.5 nM (*Nature* 2020, **584**:120); CV30, 3.6 nM (*Nat Commun* 2020 11:5413). We have modified the sentence as the following:

“...the weak pre-incubation effect by the control mAbs was probably due to residual antibodies that remained bound with pp.”

- Line 391: how is synergy determined

As shown in **Fig. EV3F**, a mild synergistic effect of CV30 and FD20 was noticed by comparing the curve and IC₅₀ of CV30+Y112R (14.3 nM, red curve) and CV30+FD20 (8.0 nM, black curve). This is measured using the black curve that was slightly shifted to the left following increased neutralization of the cocktail which provided a decrease in IC₅₀ (8.0 nM). This result indicates first that the FD20 and CV30 are compatible and do not exclude each other, and, second, that the mixture of FD20+CV30 can help CV30 arrive at a better neutralization level, even if moderate”.

We have added the following sentence to the caption of Fig. EV3:

“The slightly higher neutralizing activity of CV30+FD30 compared with CV30+Y112R indicates modest synergy of the two mAbs (Y112R is a null mutant of FD20).”

- Lines 398-411.

It appears that S-2P protein was used. This should be explicitly used instead of S whenever appropriate.

It was indeed S-2P. Changes are made to reflect this.

- Throughout the manuscript there are word choices and sentence structure formats that cause the reader to have to infer or guess what is meant. Also, there are uses of words that are not usual, like iconic, that are not typical for scientific writing and may mislead the reader to either think the writers are making claims that I do not think they mean to. Another example on Line

256 uses the phrase "could inhibit" sounds like a hypothesis, but I think it is a statement. Therefore, the manuscript would benefit from being reviewed and edited by a professional English language scientific editor or someone similar who can provide the words and sentences that convey the authors meaning with more clarity.

We have carefully revised our manuscript to improve the writing.

We thank the reviewer for the constructive suggestions to avoid unintended misleading.

References

1. Wrapp, D. et al. Cryo-EM structure of the 2019-nCoV spike in the prefusion conformation. *Science* 367, 1260-1263 (2020).
2. Hsieh, C. et al. Structure-based design of prefusion-stabilized SARS-CoV-2 spikes. *Science* 369, 1501-1505 (2020).
3. Barnes, C. O. et al. SARS-CoV-2 neutralizing antibody structures inform therapeutic strategies. *Nature* 588, 682-687 (2020).

We thank the reviewer for providing the references.

6th Oct 2021

Dear Dr. Li,

Thank you for the submission of your manuscript to EMBO Molecular Medicine. I am pleased to inform you that we will be able to accept your manuscript pending the following final amendments:

- 1) Please address all referee #3 comments.
- 2) We note that you currently have together with you, a total of 4 co-corresponding authors. Is that correct? Do you confirm equal contribution of these 4 people, able to take full responsibility for the paper and its content? While there is no limit per se to the number of co-corresponding authors, 3 is rare, 4 even more so, and may not reflect as intended to the community.
- 3) In the main manuscript file, please do the following:
 - Make sure that all special characters display well.
 - In M&M, a statistical paragraph should reflect all information that you have filled in the Authors Checklist, especially regarding randomization, blinding, replication.
 - In M&M, please specify the biosafety level for the experiments with SARS-CoV-2 by adding and amending the following sentence: All experiments with SARS-CoV-2 were performed in a ... level laboratory and with approval from...
 - In M&M, please include statement that the informed consent was obtained from all human subjects and that the experiments conformed to the principles set out in the WMA Declaration of Helsinki and the Department of Health and Human Services Belmont Report.
 - In addition to the accession number please provide URL all deposited datasets. Please be aware that all datasets should be made freely available upon acceptance, without restriction. Use the following format to report the accession number of your data:

[data type]: [full name of the resource] [accession number/identifier] ([doi or URL or identifiers.org/DATABASE:ACCESSION])

Please check "Author Guidelines" for more information.

<https://www.embopress.org/page/journal/17574684/authorguide#availabilityofpublishedmaterial>

4) Funding: Please make sure that information about all sources of funding are complete in both our submission system and in the manuscript.

5) The Paper Explained: Please provide "The Paper Explained" and add it to the main manuscript text. Please check "Author Guidelines" for more information. <https://www.embopress.org/page/journal/17574684/authorguide#researcharticleguide>

6) Synopsis:

- Synopsis text: Please provide a short stand first (maximum of 300 characters, including space) as well as 2-5 one sentence bullet points that summarise the paper as a .doc file. Please write the bullet points to summarise the key NEW findings. They should be designed to be complementary to the abstract - i.e. not repeat the same text. We encourage inclusion of key acronyms and quantitative information (maximum of 30 words / bullet point). Please use the passive voice.

- Please check your synopsis text and image, revise them if necessary and submit their final versions with your revised manuscript. Please be aware that in the proof stage minor corrections only are allowed (e.g., typos).

7) For more information: There is space at the end of each article to list relevant web links for further consultation by our readers. Could you identify some relevant ones and provide such information as well? Some examples are patient associations, relevant databases, OMIM/proteins/genes links, author's websites, etc...

8) Source data: Please upload one file per figure. For blots or microscopy, uncropped images should be submitted (using a zip archive if multiple images need to be supplied for one panel). Please check "Author Guidelines" for more information.

<https://www.embopress.org/page/journal/17574684/authorguide#sourcedata>

9) Press release: Please inform us as soon as possible and latest at the time of submission of the revised manuscript if you plan a press release for your article so that our publisher could coordinate publication accordingly.

10) Please be aware that we use a unique publishing workflow for COVID-19 papers: a non-typeset PDF of the accepted manuscript is published as "Just Accepted" on our website. With respect to a possible press release, we have the option to not post the "Just Accepted" version if you prefer to wait with the press release for the typeset version. Please let us know whether you agree to publication of a "Just accepted" version or you prefer to wait for the typeset version.

11) As part of the EMBO Publications transparent editorial process initiative (see our Editorial at <http://embomolmed.embopress.org/content/2/9/329>), EMBO Molecular Medicine will publish online a Review Process File (RPF) to accompany accepted manuscripts. This file will be published in conjunction with your paper and will include the anonymous referee reports, your point-by-point response and all pertinent correspondence relating to the manuscript. Let us know whether you agree with the publication of the RPF and as here, if you want to remove or not any figures from it prior to publication. Please note that the Authors checklist will be published at the end of the RPF.

12) Please provide a point-by-point letter INCLUDING my comments as well as the reviewer's reports and your detailed responses (as Word file).

I look forward to reading a new revised version of your manuscript as soon as possible.

Yours sincerely,

Zeljko Durdevic

***** Reviewer's comments *****

Referee #2 (Remarks for Author)

Is suitable for publication.

Referee #3 (Comments on Novelty/Model System for Author):

The manuscript is highly improved and the vast majority of this reviewers concerns have been addressed. However, unfortunately the antibody's potency and breadth are lacking and the findings within the report do not represent a high level of novelty or impact.

There are still some issues if overreach but they have mostly been addressed. For example in the abstract, the claims for showing little to no escaping effect implies that escape experiments were performed. Instead it was an analysis of virus variants.

Other comments:

- 1) Figure 1C. The curve for the SARS1 protein kinetics show an upward shift near the transition point suggesting a complex binding interaction. This could indicate an issue with the loading and could lead to an incorrect value for the KD.
- 2) When discussing Figure 4E and the conservation of epitope residues it is not clearly written and can be confusing about whether it is the full RBD or RBD that is being referred to.
- 3) The supporting data for the new Table 2 should be provided

Responses to Reviewer Comments.

Referee #2 (Remarks for Author)

Is suitable for publication.

We thank the reviewer the supportive comments.

Referee #3 (Comments on Novelty/Model System for Author):

The manuscript is highly improved and the vast majority of this reviewers concerns have been addressed. However, unfortunately the antibody's potency and breadth are lacking and the findings within the report do not represent a high level of novelty or impact.

We thank the reviewer for the constructive comments. We hope to improve the antibody's efficacy in follow-up studies.

There are still some issues if overreach but they have mostly been addressed. For example in the abstract, the claims for showing little to no escaping effect implies that escape experiments were performed. Instead it was an analysis of virus variants.

We have checked the manuscript thoroughly and re-phrased relevant texts to avoid misunderstanding as pointed by the reviewer.

In the Abstract, the sentence has been revised as:

“Mutation of the residues of the conserved epitope variably affects FD20-binding activity but confers little or no resistance to neutralization.”

In the Results, the phrase ‘the two most escaping mutants were D428G and K462A’ were changed to ‘the two most FD20-resisting mutants were D428G and K462A’.

Other comments:

1) Figure 1C. The curve for the SARS1 protein kinetics show an upward shift near the transition point suggesting a complex binding interaction. This could indicate an issue with the loading and could lead to an incorrect value for the K_D .

We repeated the experiment by changing RBD loading, the buffer compositions, and by swapping the immobilized and the analyte. However, the upward shift persists. We suspect that the commercial SARS-CoV RBD may have been partially denatured, causing the abnormal binding profile.

We have modified the relevant texts as the following:

“...Interestingly, scFD20 also bound with RBD from the closely related SARS-CoV (**Fig 1C**).”

And we added a note in the caption of **Fig. 1C**:

“Note, although the fitting of the binding curves in **D** gave a K_D of 4.8 nM, the accuracy of the kinetic parameters may be compromised owing to the mismatch between the raw data and the fitted curves at the initial dissociation phase. The reason for the abnormal BIL profile is unknown but it may be caused by possible denaturation of the commercial SARS-CoV RBD during storage. It was also noted that the BLI signal in **D** was ~5 times less than that with SARS-CoV-2 RBD under similar conditions (**C**). ”

For your information, we have attached the results of the abovementioned experiments as below (**Fig. R1**, for review purposes only).

Fig. R1. The abnormal binding profile between scFD20 and SARS-CoV RBD under various conditions. (A) The binding profile with RBD immobilized and scFD20 as analyte. The difference between this experiment and the previously reported was that the loading of RBD in **A** was reduced to a half before association. **(B)** The repeat of the experiment as in the previous **Fig. 1C** but with lower concentration of analyte (scFD20). **(C)** The binding profile with scFD20 immobilized and RBD as analyte. **(D)** Attempts to eliminate the upward shift by performing the assay in different buffers with detergents, salts, and BSA as additives. Buffer conditions are listed as in **a-e**. **(E)** A positive control (SARS-CoV-2 RBD immobilized and a nanobody as analyte) to demonstrate the performance of the machine at a similar BLI signal level.

2) When discussing Figure 4E and the conservation of epitope residues it is not clearly written and can be confusing about whether it is the full RBD or RBD that is being referred to.

We have added a note to clarify this point.

“of which the RBD is 89.5% identical and 94.0% similar to SARS-CoV-2 RBD (**Fig. 4E**) (note, residues Pro330 and Lys529 of SARS-CoV-2 S was used to calculate sequence identity/similarity).”

3) The supporting data for the new Table 2 should be provided

Done

We are pleased to inform you that your manuscript is accepted for publication and is now being sent to our publisher to be included in the next available issue of EMBO Molecular Medicine.

Corresponding Author Name: Junfang Chen, Fei-Long Meng, Dimitri Lavillette, Dianfan Li

Manuscript Number: EMM-2021-14544